# Prune 'n Predict: Optimizing LLM Decision-making with Conformal Prediction

**Harit Vishwakarma** [1 2]  **Alan Mishler** [1]  **Thomas Cook** [1]  **Niccolò Dalmasso** [1]  **Natraj Raman** [1]  **Sumitra Ganesh** [1]

## Abstract

Large language models (LLMs) are empowering decision-making in several applications, including tool or API usage and answering multiple-choice questions (MCQs). However, incorrect outputs pose significant risks in high-stakes domains like healthcare and finance. To quantify LLM uncertainty and thereby mitigate these risks, recent works employ conformal prediction (CP), a model- and distribution-agnostic framework that uses LLM outputs to generate a *prediction set* containing the true answer with high probability. Leveraging CP, we propose *conformal revision of questions* (CROQ), which revises the question by narrowing down the available choices to those in the prediction set and asking the LLM the revised question. We expect LLMs to be more accurate on revised questions with fewer choices. Furthermore, we expect CROQ to be effective when the prediction sets from CP are small. Commonly used logit scores often lead to large sets, diminishing CROQ's effectiveness. To overcome this, we propose CP-OPT, an optimization framework to learn scores that minimize set sizes while maintaining coverage. Our extensive experiments on MMLU, ToolAlpaca, and TruthfulQA datasets with multiple LLMs show that CROQ improves accuracy over the standard inference, with more pronounced gains when paired with CP-OPT.

## 1. Introduction

Large language models (LLMs) (Touvron et al., 2023; Databricks, 2024; Abdin et al., 2024) have demonstrated remarkable capabilities in various decision-making tasks, including multi-choice question answering and tool usage,

[1]JPMorganChase AI Research, New York, NY, USA [2]Department of Computer Science, University of Wisconsin, Madison, WI 53706, USA. This work was performed while at JP-MorganChase. Correspondence to: Harit Vishwakarma <hvishwakarma@cs.wisc.edu>.

*Proceedings of the 42^{nd} International Conference on Machine Learning*, Vancouver, Canada. PMLR 267, 2025. Copyright 2025 by the author(s).

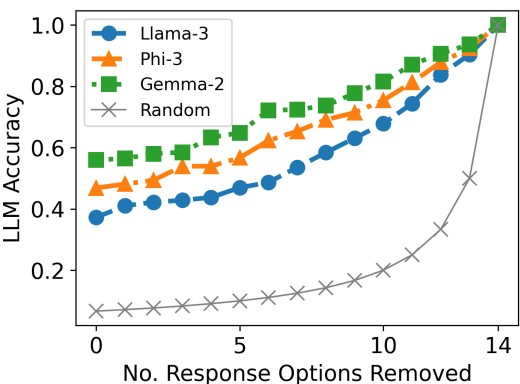

Figure 1: Accuracy for three LLMs on the TruthfulQA dataset with 15 response options as a function of the number of incorrect answer options (distractors) removed from the prompt. As more distractor answers are eliminated, accuracy increases. Accuracy is averaged across 5 iterations, error bars denote $\pm 2$ standard deviations.

where the model must select the correct tool or API to complete a task (Qu et al., 2024; Tang et al., 2023; Hendrycks et al., 2021). However, LLMs often exhibit overconfidence in wrong answers (Krause et al., 2023; Groot & Valdenegro Toro, 2024). Such unreliable predictions entail significant risks in critical domains like finance. Successful usage in such settings demands principled solutions to improve accuracy and quantify uncertainty in the predictions.

A commonly taught strategy for human test takers to solve multi-choice questions (MCQs) is the process of elimination (pruning) of incorrect (distractor) answer choices. The underlying principle is that this enables them to focus their attention on the remaining answer choices, and it increases the likelihood of a correct answer even if they have to guess randomly. Inspired by this, we investigate whether LLMs can benefit from a similar strategy.

We first examine the relationship between the number of distractor answers and LLM accuracy on an MCQ task. Figure 1 illustrates accuracy for three different LLMs on a version of TruthfulQA, a widely used MCQ dataset. The MCQs in this version of TruthfulQA have 15 answer options, only one of which is correct. (We discuss how this dataset is constructed in Appendix E.2.) For each question, we repeatedly prompt the LLM, randomly eliminating one distractor

answer at a time. Each prompt is independent, without any previous rounds included in the context. As hypothesized, *reducing the number of response options leads to an improvement in accuracy*, and this improvement is very nearly monotone. This suggests that eliminating distractor answers before prompting the LLM can indeed enhance accuracy. Of course, when pruning answers, we do not want to eliminate the correct answer, since that would necessarily cause the LLM to get the MCQ wrong.

Conformal prediction (CP) (Vovk et al., 2005) is a flexible framework that can be used to prune distractor answers while retaining the correct answer with high probability. CP is a *model-agnostic* and *distribution-free* technique for generating prediction sets which contain the correct answer with a user-specified probability (e.g., 95%), which is referred to as the *coverage guarantee*.

Utilizing this guarantee of CP, we propose a procedure called *conformal revision of questions* (CROQ), to revise MCQs with choices in a prediction set output by CP. This procedure represents a tradeoff: with some small probability (e.g., 5%), we may remove the correct answer from the prediction set, causing the LLM to get the question wrong. However, with high probability (e.g., 95%), we will retain the correct answer while reducing the number of distractor answers. Given the relationship observed in Figure 1, this should increase the LLM's accuracy on those questions. Different coverage rates naturally induce different tradeoffs. Overall, we hypothesize that we can find a coverage rate with a favorable tradeoff, such that CROQ improves the overall accuracy on a given MCQ task.

Figure 1 suggests that CROQ's effectiveness should depend on the size of the prediction sets from conformal prediction – smaller sets mean fewer choices in the revised question and hence better final accuracy. Conformal prediction requires specifying a *score function*, which loosely speaking quantifies how plausible an output (answer option) is with respect to a given input (question). While conformal prediction provides a coverage guarantee for *any* score function, the size of the prediction sets depends on the score function. As an example, a random score function will yield output sets that constitute random subsets of the label space that are large enough to satisfy the coverage guarantee (Angelopoulos & Bates, 2022).

Previous works that apply conformal prediction in MCQ-type settings have used readily available scores such as the logits (or softmax values) output from the LLM (Kumar et al., 2023) or have designed heuristic scores based, for example, on repeated querying of the LLM (Su et al., 2024). Logits can be overconfident and may show biases for some options (Zheng et al., 2024), and heuristic scores are not guaranteed to produce small sets. Thus, in order to make CROQ as effective as possible, we propose CP-OPT (confor-

mal prediction optimization), a principled solution to obtain scores that are designed to minimize set sizes (uncertainty) while preserving the coverage guarantee.

To summarize, our main contributions are as follows:

1. We propose the conformal revision of questions (CROQ), in which we prune the answer choices in an MCQ to those in the prediction set output by conformal prediction and then prompt the LLM with the revised question. Empirical evaluation shows that this approach consistently improves accuracy compared to prompting the LLM with the original MCQ.

2. We design a score function optimization framework (CP-OPT) that can be applied to any pre-trained LLM. Moving away from the potentially unreliable LLM logits and heuristic scores, our framework provides a principled way to learn scores for conformal prediction. Empirically, we show that our procedure leads to a reduction in average set sizes compared to the baseline procedure that uses the LLM logits as the scores, at the same level (95%) of coverage.

3. We further show that when used with CROQ, our CP-OPT scores deliver greater accuracy improvements over baseline than the LLM's logits.

## 2. Preliminaries

In this section, we provide background on solving MCQ tasks with LLMs and conformal prediction.

### 2.1. Multiple Choice Questions (MCQs) and LLMs

**MCQ Setup.** MCQs are a general abstraction for expressing problems in which the correct choice(s) must be selected from a given set of choices. These encompass question-answering tasks like MMLU (Hendrycks et al., 2021) as well as other tasks such as tool learning, in which the LLM must select the correct tool or API to complete a task (Tang et al., 2023; Qu et al., 2024). An MCQ consists of the question text $Q$, i.e. a sequence of tokens, and a set of answer choices $O = \{(Y_1, V_1), (Y_2, V_2), \ldots, (Y_m, V_m)\}$. Here, each $Y_j$ is a unique character from the English alphabet, and we assume that the number of choices $m$ is less than or equal to the size of the alphabet. Each $V_j$ is the option text for the $j$th option. Denote the whole MCQ instance as $x = (Q, O)$. Let $\mathcal{X}_m$ denote the space of MCQs with $m$ choices and $\mathbb{P}_{\mathcal{X}_m}$ denote a distribution over $\mathcal{X}_m$, from which samples for training, calibration, and testing are drawn independently. Here, we assume that for each question $Q$ there is only one correct answer key $y^\star \in \{Y_1, Y_2, \ldots Y_m\} = \mathcal{Y}_m$.

**MCQ Prompt.** We concatenate the question text $Q$ and

the answer choices $O$, all separated by a new line character, and append to the end the text "The correct answer is: ". The expectation is that given this input prompt, the next token predicted by the LLM will be one of the option keys. See Appendix E for a prompt example. We consider zero-shot prompts and do not include example questions and answers in the prompt. We also add the prefix and suffix tokens to the prompt as recommended by the language model providers. Since these are fixed modifications to $x$, we will use $x$ to denote the final prompt and the MCQ instance analogously.

**LLM Inference.** We run the forward pass of the auto-regressive LLM (Touvron et al., 2023; Dubey et al., 2024; Abdin et al., 2024) on the input prompt to obtain the logit scores for each possible next token given the prompt, restricting attention to the tokens that correspond to the available answer keys (e.g. "a", "b", "c", "d" if there are four answer options). We take the softmax to convert the logits to probabilities, and then we take as the LLM's answer the option with the highest probability. This approach ensures that the LLM's answer will be one of the available answer options, which would not be guaranteed if instead we asked the LLM to simply generate an answer token given the prompt. This approach mirrors what has been done in other works that use LLMs to solve MCQs (Kumar et al., 2023; Su et al., 2024). Formal details are given in Appendix A.1.

## 2.2. Conformal Prediction

Conformal prediction (CP) (Vovk et al., 2005; Angelopoulos et al., 2022) is a framework for quantifying uncertainty in machine learning models. It provides a flexible and user-friendly approach to output *prediction sets* (which may be finite sets or intervals) that contain the true output or label with a probability that is specified by the user, e.g. 95%. The key strength of conformal prediction lies in its *distribution-free* guarantees: it ensures that the constructed prediction sets are valid regardless of the underlying data distribution and model. This property is particularly desirable in the context of language models, as it is hard to characterize language data distributions or put specific distributional assumptions/restrictions on the LLMs.

**Score Function.** Let $g : \mathcal{X}_m \times \mathcal{Y}_m \mapsto \mathbb{R}$ be a conformal *score function*, where larger scores indicate better agreement ("conformity") between $x$ and $y$. Intuitively, large scores are intended to indicate that $y$ is a plausible output given $x$, while smaller scores indicate less plausibility. (Note that some authors prefer to have larger scores indicate greater disagreement, e.g. Clarkson et al. (2024).) A common choice of score function is the softmax scores from the given model. For closed-source LLMs, where logits are not available, others have devised self-consistency scores based on repeated querying of the model (Su et al., 2024).

**Prediction Sets.** Given a score function $g$ and threshold $\tau$ on the scores, the prediction set for any $x \in \mathcal{X}_m$ is given by

$$C(x; g, \tau) := \{y \in \mathcal{Y}_m : g(x, y) \geq \tau\}. \quad (1)$$

Intuitively, larger sets represent greater uncertainty, while smaller sets represent less uncertainty. Given a fixed confidence level, a score function that produces larger sets can be said to result in greater uncertainty.

**Split Conformal Prediction.** Similar to prior works (Kumar et al., 2023; Su et al., 2024), we use *Split Conformal Prediction* (Papadopoulos et al., 2002; Lei et al., 2018) due to its popularity, ease of use, and computational efficiency. Given a score function $g : \mathcal{X}_m \times \mathcal{Y}_m \mapsto \mathbb{R}$, Split Conformal Prediction uses a calibration dataset $D_{\text{cal}} = \{x_i, y_i^\star\}_{i=1}^{n_{\text{cal}}}$ to compute a threshold $\widehat{\tau}_\alpha$, defined as

$$\hat{\tau}_\alpha := \inf \left\{ q : \widehat{F}_g(q) \geq \frac{\lfloor (n_{\text{cal}} + 1)\alpha \rfloor}{n_{\text{cal}}} \right\}, \quad (2)$$

where, $\widehat{F}_g(q) := \frac{1}{n_{\text{cal}}} \sum_{i=1}^{n_{\text{cal}}} \mathbb{1}\left(g(x_i, y_i^\star) \leq q\right)$ is the empirical CDF (cumulative distribution function) of scores from $g$ and $\alpha \in [0, 1]$ is a user-chosen *miscoverage rate* that is equal to 1 minus the desired coverage; for example, a value of $\alpha = 0.05$ would correspond to a coverage of 95%. In words, $\hat{\tau}_\alpha$ is the smallest empirical quantile of the scores for the correct answers on the calibration dataset that is sufficient to satisfy (an empirical version of) the coverage property. The threshold $\hat{\tau}_\alpha$ is used to construct prediction sets $C(x; g, \hat{\tau}_\alpha)$ on previously unseen test points as in (1). This procedure enjoys a marginal coverage guarantee for prediction sets on unseen test data points, formalized as Proposition 2.1. A proof is provided in Appendix B.1.

**Proposition 2.1.** *(Marginal Coverage Guarantee) Let $g$ be a fixed conformity score function and $\hat{\tau}_\alpha$ be an $\alpha$ threshold computed via Split Conformal Prediction on $D_{\text{cal}} = \{x_i, y_i^\star\}_{i=1}^{n_{\text{cal}}} \overset{iid}{\sim} \mathbb{P}_{\mathcal{X}_m \times \mathcal{Y}_m}$. Then, for a new sample $(\tilde{x}, \tilde{y}^\star) \sim \mathbb{P}_{\mathcal{X}_m \times \mathcal{Y}_m}$, we have that*

$$\mathbb{P}(\tilde{y}^\star \in C(\tilde{x}; g, \hat{\tau}_\alpha)) \geq 1 - \alpha. \quad (3)$$

*where the probability is marginal over the randomness in the calibration data and the new sample.*

The top half of Figure 2 illustrates conformal prediction for answering MCQs with LLMs. While the coverage guarantee in Proposition 2.1 holds for any score function, ideally, we would like a score function that yields the smallest sets possible (the least uncertainty). Next, we discuss our solutions to improve conformal prediction and its utility in solving MCQs with LLMs.

## 3. Methodology

In this section, we discuss details of our pipeline for question revision using conformal prediction and our procedure to generate optimal conformal scores.

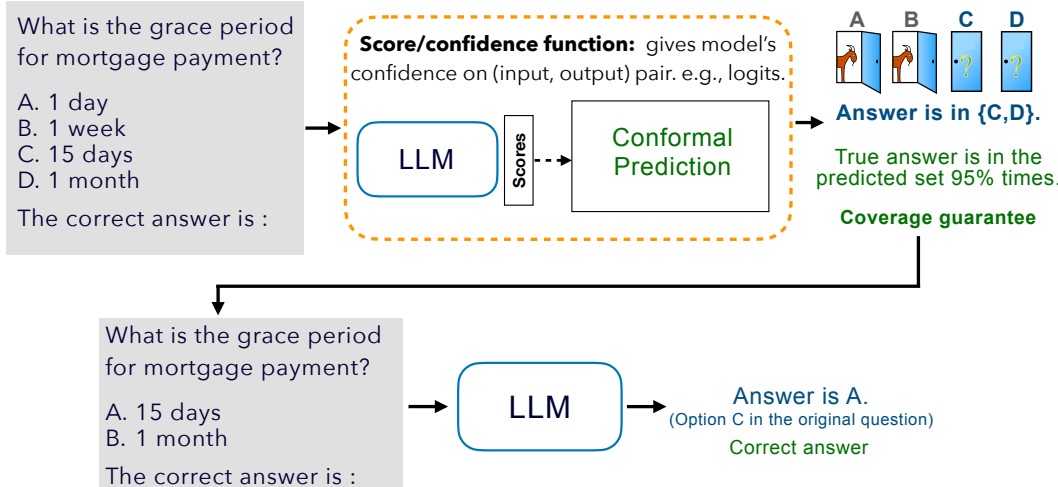

Figure 2: (CROQ) Illustration of conformal revision of questions and prompting the LLM with the revised question. In this example, the initial predicted set by LLM + conformal prediction (CP) is {C, D}. The question and labels are revised to contain only the answer choices in the prediction set and the LLM is prompted with the revised question. Since CP provides rigorous coverage guarantees, we expect that re-prompting the LLM with reduced answer choices will improve the chances of obtaining the correct answer. See Section 3.1 for more details.

### 3.1. Conformal Revision of Questions (CROQ)

The procedure involves prompting the LLM with the reduced answer options from a conformal prediction set. The steps are illustrated with an example in Figure 2.

**Scores and Threshold for Conformal Prediction.** We first fix a score function $g : \mathcal{X}_m \times \mathcal{Y}_m \mapsto \mathbb{R}$. Here we restrict the score function to either the logits generated by the LLM or the CP-OPT scores discussed in Section 3.2. We then run the split conformal procedure with coverage level $1 - \alpha$ for some $\alpha \in [0, 1]$ to estimate the threshold $\hat{\tau}_\alpha$. CROQ then proceeds as follows.

**Step 1: Get Conformal Prediction Set.** Given a test instance $x$, we generate a first stage prediction set, $C(x\,;\,g, \hat{\tau}_\alpha)$. Per the coverage guarantee (Proposition 2.1), we expect that the true answer $y^\star \in C(x\,;\,g, \hat{\tau}_\alpha)$ with probability at least $1 - \alpha$. Next, the question is revised to contain only the choices in the set $C(x\,;\,g, \hat{\tau}_\alpha)$.

**Step 2: Revise the Question and Ask the LLM.** If the first stage prediction set $C(x\,;\,g, \hat{\tau}_\alpha)$ is empty or is of size 1 or size $m$ (the number of answer options), then we simply utilize the LLM's answer to the original MCQ $x$, as described in section 2.1, since the conformal procedure has yielded no additional information. Otherwise, we modify the prompt $x$ to $x' = (Q, O')$, where $O' = \{(K_j, V_j) : K_j \in C(x\,;\,g, \hat{\tau}_\alpha)\}$. The keys in $O'$ are changed so that they start with the first letter of the alphabet and go to the letter corresponding to the number of choices available. For example, if there were initially four answer options {a, b, c, d}, and the conformal prediction set was {c, d}, then the

two options in the set would receive new keys {a, b}. Then $x'$ is transformed into a prompt format and passed to the LLM, and the standard inference procedure (section 2.1) is run to extract the predicted answer key $\hat{y}'$.

With fewer choices in the revised question, we expect LLMs will be more accurate in their answer compared to the answer to the initial question. However, we also expect that the improvement in accuracy will depend on the size of the prediction sets, as illustrated in Figure 1. We provide a simple analysis to elaborate this point.

**Characterizing Accuracy Improvements.** Let $a := \mathbb{P}(\hat{y} = y^\star \mid x \in \mathcal{X}_m)$ denote the accuracy of standard single-round inference from an LLM (without CROQ) on questions with $m$ answer options. By pruning answer choices with conformal prediction (the first step of CROQ), we obtain modified questions. We can group these modified questions by the number of remaining answer options. Let $\nu(x) := |C(x;g,\hat{\tau}_\alpha)|$ denote the size of the prediction set (the number of answer options after pruning) for question $x$. Let $r_k := \mathbb{P}(\nu(x) = k)$ denote the proportion of questions having $k$ options after pruning, for $k = 1, \ldots, m$. We have $\sum_{k=1}^{m} r_k \leq 1$. (We exclude sets of size 0 because these necessarily do not contain the correct answer.) The coverage on this set of questions is $\rho_k := \mathbb{P}(y^\star \in C(x;g,\hat{\tau}_\alpha) \mid \nu(x) = k) = 1 - \alpha_k$, for some $\alpha_k \in [0, 1]$. It is easy to see that $\sum_{k=1}^{m} r_k \rho_k \geq 1 - \alpha$, due to Proposition 2.1. In other words, $\alpha_k$ have to be such that $\sum_{k=1}^{m} r_k \alpha_k \leq \alpha$.

Let $f_{\text{post}}(k) := \mathbb{P}(\hat{y} = y^\star \mid \nu(x) = k, y^\star \in C(x;g,\hat{\tau}_\alpha))$

denote the accuracy of the LLM on questions with $k$ answer options after CROQ has been applied, when the correct answer is present in the prediction set. The monotonicity observed in Figure 1 suggests that it is reasonable to expect $f_{\text{post}}(k)$ to be monotone in $k$, i.e., as the number of options increases, the accuracy may decrease.

**Assumption 3.1.** *(Monotone Accuracy) The conditional accuracy function $f_{post}(k)$ is monotonically decreasing in $k$.*

**Proposition 3.2.** *Given the definitions above, the following statements hold.*

1. *The change in accuracy due to CROQ is given as $\Delta(f_{post}, \alpha, a) := \sum_{k=1}^{m} r_k \rho_k f_{post}(k) - a$.*

2. *A sufficient condition for positive gain $\Delta(f_{post}, \alpha, a) > 0$ is $r_k \rho_k > \frac{a}{m f_{post}(k)}$ for all $1 \leq k \leq m$.*

3. *Suppose that the accuracy function $f_{post}(k)$ is fixed and that it satisfies Assumption 3.1. Then among all possible sets of pairs $\{(r_k, \rho_k)\}_{k=1}^{m}$ satisfying $\sum_{k=1}^{m} r_k \leq 1$ and $1 - \alpha \leq \sum_{k=1}^{m} r_k \rho_k \leq 1$, the gain $\Delta(f_{post}, \alpha, a)$ is maximized by the greedy solution that sets $r_1 \rho_1$ as large as possible, then sets $r_2 \rho_2$ as large as possible, etc., such that $r_1 \rho_1 \geq r_2 \rho_2 \geq \ldots \geq r_m \rho_m$.*

Proof is given in Appendix B.2. This proposition illustrates the interplay between coverage and accuracy at different set sizes. Claim 3 suggests that to maximize the gain in accuracy, a high proportion $r_k$ and coverage $\rho_k$ for smaller $k$ (set size) is preferable.

The split CP procedure lacks direct control on $r_k$, $\rho_k$, and $f_{\text{post}}(k)$; instead, it tunes a threshold $\hat{\tau}_\alpha$ for any score function $g$, such that the marginal coverage of the sets $C(x; g, \hat{\tau}_\alpha)$ is at least $1 - \alpha$. If we can minimize the set sizes while maintaining the coverage guarantee, we can indirectly increase $r_k$, $\rho_k$ for smaller values of $k$ and in turn obtain higher accuracy gains.

While conformal prediction can output sets $C(x; g, \hat{\tau}_\alpha)$ for any score function $g$, along with $1 - \alpha$ coverage guarantee, the set sizes could be highly variable depending on the score function $g$. Noting the lack of reliability of scores used in prior works, that could yield unnecessarily large sets, we seek to learn scores that minimize the set sizes while preserving the coverage guarantee. We discuss our procedure to learn such scores in the next section. Using these scores in CP, we expect to get smaller sets and thus more improvement in CROQ compared to baseline scores.

The simple analysis above provides insights into how CROQ can lead to improvements in accuracy. We believe the monotonicity property expressed in Assumption 3.1 is a useful point of departure for more in-depth analyses, which we leave for future work. We conclude this section with some closing remarks on the CROQ procedure.

*Remark* 3.3. The score function used to prune the answer choices in Step 1 of CROQ can come from any source, including a different LLM from the one used in Step 2 or a method that does not require querying an LLM. This flexibility enables combining knowledge from multiple LLMs, and it can be useful for example when the number of options is large, resulting in costly LLM inference in the first round. In such settings, cheaper alternatives like pairwise similarities can be used to prune the choices. We illustrate the benefits of this flexibility empirically in the NL2SQL use case in Appendix C.3.

*Remark* 3.4. Our proposed CROQ procedure is limited to two steps, but in principle, CROQ can be run over multiple rounds. Each round will successively prune the answer choices until the last round yields a final answer. This simple extension to multi-round CROQ may yield better results, but there are a few challenges that have to be addressed to make it practical. First, the computational cost increases with the number of rounds (though, as discussed in the previous remark, cheap scoring procedures may keep this cost low). Second, the conformal procedure in each round has to be calibrated for a higher coverage than $1 - \alpha$ so that the eventual coverage of the prediction sets in the penultimate round is at least $1 - \alpha$. Lastly, using the same calibration data in each round can introduce biases and make the coverage guarantee invalid. We believe these challenges can be overcome with a larger calibration set, more compute time, and careful selection of coverage parameters for each round. Studying this will be a fruitful direction for future work.

### 3.2. CP-OPT to Optimize Scores

We describe our method for learning the optimal scores for conformal prediction (CP) for solving MCQs with LLMs. Similar ideas have been incorporated in the training objective of classifiers (Stutz et al., 2022) so that the classifiers' softmax output is better suited for CP. However, the LLMs are not trained with this objective, and we want to apply CP to any given LLM; therefore, we design a post-hoc method to optimize the scores. We first characterize the optimal scores and then describe how to estimate them in practice.

**Characterization of the optimal scores.** For any score function $g : \mathcal{X}_m \times \mathcal{Y}_m \mapsto \mathbb{R}$ and threshold $\tau$, the membership of any $y$ in the prediction set $C(x; g, \tau)$ is given by $\mathbb{1}(y \in C(x; g, \tau)) = \mathbb{1}\{g(x, y) \geq \tau\}$. Define the expected set size $S(g, \tau)$ and the coverage conditional on $\tau$, denoted $\mathcal{P}(g, \tau)$, as follows:

$$S(g, \tau) := \mathbb{E}_x \Big[ \sum_{y \in \mathcal{Y}_m} \mathbb{1}\{g(x, y) \geq \tau\} \Big]. \qquad (4)$$

$$\mathcal{P}(g, \tau) := \mathbb{E}_x \left[ \mathbb{1}\{g(x, y^\star) \geq \tau\} \right]. \qquad (5)$$

The optimal score function $g^\star$ and threshold $\tau^\star$ are defined (non-uniquely) to minimize the expected set size subject to the coverage $\mathcal{P}(g, \tau)$ being at least $1 - \alpha$:

$$g^\star, \tau^\star := \underset{g:\mathcal{X}_m \times \mathcal{Y}_m \mapsto \mathbb{R}, \tau \in \mathbb{R}}{\arg\min} S(g, \tau) \text{ s.t. } \mathcal{P}(g, \tau) \geq 1-\alpha. \tag{P1}$$

**Practical Version with Differentiable Surrogates and Empirical Estimates.** Problem (P1) characterizes optimal score functions and thresholds. However, in practice, we do not know the underlying distribution and thus do not have access to the quantities in (4) and (5). Instead, we obtain their estimates using a training sample $D_{\text{train}} = \{(x_i, y_i^\star)\}_{i=1}^{n_t}$ drawn independently from the same distribution:

$$\widehat{S}(g, \tau) := \frac{1}{n_t} \sum_{i=1}^{n_t} \sum_{y \in \mathcal{Y}_m} \mathbb{1}\{g(x_i, y) \geq \tau\}, \tag{6}$$

$$\widehat{\mathcal{P}}(g, \tau) := \frac{1}{n_t} \sum_{i=1}^{n_t} \mathbb{1}\{g(x_i, y_i^\star) \geq \tau\}. \tag{7}$$

Using these plug-in estimators in problem (P1) yields a revised optimization problem. However, it is difficult to solve this problem as the objective and constraints are not differentiable. To make them differentiable, we introduce the following surrogates. Given $g(x, y)$ and $\tau$, define the following sigmoid function with $\beta > 0$, $\sigma(x, y, g, \tau, \beta) := 1/\big(1 + \exp(-\beta\,(g(x, y) - \tau))\big)$. The sigmoid function provides a differentiable approximation to the indicator variable for $g(x, y) \geq \tau$. The approximation is tighter with larger $\beta$ i.e., $\sigma(x, y, g, \tau, \beta) \to \mathbb{1}\{g(x, y) \geq \tau\}$ as $\beta \to \infty$, and $g(x, y) \geq \tau \iff \sigma(x, y, g, \tau) \geq 1/2$. By using these sigmoid surrogates in equation (6), we obtain the following smooth plugin estimates,

$$\widetilde{S}(g, \tau) := \frac{1}{n_t} \sum_{i=1}^{n_t} \sum_{y \in \mathcal{Y}_m} \sigma(x_i, y, g, \tau, \beta). \tag{8}$$

$$\widetilde{\mathcal{P}}(g, \tau) := \frac{1}{n_t} \sum_{i=1}^{n_t} \sigma(x_i, y_i^\star, g, \tau, \beta). \tag{9}$$

It is easy to see that by the strong law of larger numbers and properties of the sigmoid function, as $n_t, \beta \to \infty$, the surrogate average set size and coverage will converge almost surely to their population versions, i.e. $\widetilde{S}(g, \tau) \xrightarrow{a.s.} S(g, \tau)$ and $\widetilde{\mathcal{P}}(g, \tau) \xrightarrow{a.s.} \mathcal{P}(g, \tau)$. We replace the expected set size and marginal coverage by these smooth surrogates in (P1) and transform it into an unconstrained problem with a penalty term $\lambda > 0$. We also introduce $\ell_2$ regularization to encourage low norm solutions. We optimize the score function $g$ over a flexible space of functions $\mathcal{G}$, such as neural networks (NNs). The resulting problem (P2) is

differentiable, and we solve it using stochastic gradient descent.

$$\tilde{g}, \tilde{\tau} := \underset{g \in \mathcal{G}, \tau \in \mathbb{R}}{\arg\min} \widetilde{S}(g, \tau) + \lambda\big(\widetilde{\mathcal{P}}(g, \tau) - 1 + \alpha\big)^2$$
$$- \hat{\mathcal{C}}(g) + \lambda_1 \|g\|_2^2. \tag{P2}$$

Here, $\hat{\mathcal{C}}(g) := \frac{1}{n_t} \sum_{i=1}^{n_t} \log(g(x_i, y_i^*))$ is the cross entropy term included to encourage higher scores for correct predictions, and the regularization term $\lambda_1 \|g\|_2^2$ is the squared norm over the parameters of $g$ to promote low norm solutions. Solving (P2) yields a score function $\tilde{g}$ and a threshold $\tilde{\tau}$. However, $\tilde{\tau}$ may be biased, since it is estimated on the same data as $\tilde{g}$. Following the split conformal procedure, we therefore estimate a new threshold $\hat{\tau}$ on a separate calibration dataset. Note that our framework is flexible and can work with any choice of features and function class for which the $\ell_2$ norm can be calculated. We discuss the specific choice of features and $\mathcal{G}$ used in this work.

**Specific choice of features and $\mathcal{G}$.** In practice, we want to use a flexible and easy-to-train function class for $\mathcal{G}$, as this is a post-hoc procedure and we want to avoid expensive fine-tuning. We use 3-layer neural networks with `tanh` activation as $\mathcal{G}$ and use the LLM's logits and the penultimate layer's representations corresponding to the last token as input features to the $g$ network. Let $\boldsymbol{z} \in \mathbb{R}^{d+m}$ be the concatenation of the LLM's penultimate layer's representation ($d$-dimensional) and logits ($m$-dimensional) for the last token. Our choice of $\mathcal{G}$ for the experiments is defined as follows,

$$\mathcal{G} := \{ g : \mathbb{R}^{d_0} \to \Delta^{m-1} \mid g(\boldsymbol{z}) := \texttt{softmax}(\\ \boldsymbol{W}_3 \texttt{tanh}(\boldsymbol{W}_2 \texttt{tanh}(\boldsymbol{W}_1(\boldsymbol{z})))), \\ \boldsymbol{W}_1 \in \mathbb{R}^{d_0 \times d_1}, \boldsymbol{W}_2 \in \mathbb{R}^{d_1 \times d_2}, \\ \boldsymbol{W}_3 \in \mathbb{R}^{d_2 \times m} \}$$

Here, $d_0 = d+m$, $d_1 = (d+m)/2$, and $d_3 = (d+m)/4$ and $\Delta^{m-1}$ is the $m - 1$ dimensional probability simplex. This class for $\mathcal{G}$ is flexible enough and the resulting optimization problem is not computationally prohibitive to solve. More complex (flexible) choices of $\mathcal{G}$ could be used when we can devote more compute to learning the score function.

# 4. Experiments

We conduct experiments on benchmark MCQ and tool usage tasks with open-weight instruction-tuned models to test the following hypotheses:

**H1.** CP-OPT scores in conformal prediction on MCQ tasks with LLMs yield a smaller average set size at the same level of coverage in comparison to using LLM logits.

**H2.** Conformal revision of questions (CROQ) improves accuracy over the standard inference procedure.

**H3.** CROQ with CP-OPT scores performs better than CROQ with logit scores.

### 4.1. Experimental Setup

We first describe the setup for the experiments and then discuss the results for the above hypotheses.

**Datasets.** We evaluate our hypotheses on 3 datasets: MMLU (Hendrycks et al., 2021), TruthfulQA (Lin et al., 2022), and ToolAlpaca (Tang et al., 2023). MMLU and TruthfulQA are popular benchmark datasets for multiple-choice questions. MMLU focuses on assessing multitask accuracy; it contains multiple choice questions (MCQs) from 57 domains, including humanities, math, medicine, etc. TruthfulQA evaluates an LLM's ability to answer truthfully and avoid falsehoods that humans are susceptible to. ToolAlpaca contains 3.9k tool-use instances from a multi-agent simulation environment, which we augment to a MCQ format. Dataset descriptions and example questions and responses are provided in Appendix E.

**Models.** We use auto-regressive language models based on the transformer architecture. We choose instruction-tuned, open-weight, and small to medium-sized models, for reproducibility and reduced computational cost. Specifically, we use Llama-3-8B-Instruct by Meta (Dubey et al., 2024), Phi-3-4k-mini-Instruct by Microsoft (Abdin et al., 2024), and the gemma-2-9b-it-SimPO model (Meng et al., 2024). For brevity, we use the short names Llama-3, Phi-3, and Gemma-2 respectively for these models.

**Choices of Scores.** We use the following scores for conformal prediction. (1) LLM Logits (Softmax) are extracted from the LLM as discussed in Section 2.1. These have been used in prior works (Kumar et al., 2023; Su et al., 2024). (2) CP-OPT (Ours) are the scores learned using the score optimization procedure discussed in Section 3.2. We use the train split for each dataset to learn these scores. The hyperparameter settings we used for CP-OPT are given in Appendix F. We omit the self-consistency based heuristic scores proposed by Su et al. (2024), as these require repeated inferences to get good estimates of the scores, and hence have a high computational cost.

We use the provided validation splits as our calibration datasets for the conformal procedure. For testing the hypotheses, we calibrate the conformal threshold for the coverage guarantee of 95%, i.e. we set the miscoverage rate $\alpha$ to 0.05. In addition, we study CROQ with calibration in a range of $\alpha$ values: {0.01, 0.02, 0.03, 0.04, 0.05, 0.06, 0.07, 0.08, 0.09, 0.1, 0.15, 0.2, 0.25, 0.3, 0.4, 0.5 }. Performance is computed on test splits. The hyperparameters used to learn the score function using SGD are provided in table 21 in Appendix F.

**Statistical Significance.** We report the statistical significance of our results using paired sample t-tests, using asterisks (*) to annotate results that are statistically significant at a 0.05 significance level. See Appendix D for details.

### 4.2. Discussion

***H1. Improvement in conformal set sizes with our CP-OPT scores.*** We run the CP procedure using the LLM logits and CP-OPT scores and obtain conformal sets for points in the test sets. We compute the average set size and coverage for each dataset, model, and score combination. The results are in Table 1. As expected, in most settings (17 out of 27) we see a statistically significant reduction in the set sizes with our (CP-OPT) scores with similar coverage as logits. The reduction is more pronounced with a higher number of options. In a few settings (6/27), the reduction in set size is accompanied by a statistically significant decrease in coverage relative to using the logits. In the remaining 4/27 settings the differences are insignificant. Note that since the target coverage level is 95%, anything above 95% is over-coverage. We see that logits tend to over-cover and thus a drop in coverage is expected as long as it does not fall significantly below the desired level of 95% (this happens only in 2/27 settings). Overall, these results show CP-OPT's effectiveness in reducing set sizes while maintaining the target coverage level. In Appendix C, we provide histograms (e.g., Figure 6) of set sizes produced by logits and CP-OPT scores in all settings. These histograms show a clear pattern: CP-OPT scores produce fewer large sets and more small sets in comparison to logit scores.

***H2. Accuracy improvement with conformal revision of questions (CROQ).*** Tables 2 and 5 show the accuracy before and after CROQ with logit and CP-OPT scores respectively. With the logit scores (Table 2), we see an increase in accuracy (by up to 6.43%) in 19 out of 27 settings, out of which 9 are statistically significant. In 8 of the settings, we see a small drop in accuracy (which is not statistically significant). Next, with CP-OPT scores (Table 5) we see accuracy improvements (up to 7.24%) in 24 settings, of which 13 are statistically significant. In the remaining 3 settings, we see a non-significant drop in accuracy. Overall, we observe that in the vast majority of the settings, CROQ improves accuracy with either logits or CP-OPT scores. The rare small drops in accuracy could occur since the conformal procedure may eliminate the correct option with low probability ($\alpha$).

***H3. CROQ with CP-OPT scores is better than CROQ with logit scores.*** CP-OPT scores are designed to minimize set sizes while maintaining the coverage guarantee. As a result, using these scores with CROQ is expected to reduce uncertainty for many questions, leading to fewer answer

| | | Llama-3 | | | | Phi-3 | | | | Gemma-2 | | | |
|---|---|---|---|---|---|---|---|---|---|---|---|---|---|
| | | Avg. Set Size | | Coverage | | Avg. Set Size | | Coverage | | Avg. Set Size | | Coverage | |
| Dataset | # Opt. | Logits | Ours | Logits | Ours | Logits | Ours | Logits | Ours | Logits | Ours | Logits | Ours |
| MMLU | 4 | 2.56 | **2.53*** | 95.75 | 95.57 | 2.21 | **2.16*** | 94.65 | 94.35 | 2.94 | **2.40*** | 95.16* | 94.23 |
| | 10 | 5.53 | **4.90*** | 96.06* | 95.45 | 4.36 | **4.36** | 94.11 | 94.09 | 7.79 | **6.08*** | 95.00* | 94.04 |
| | 15 | 7.69 | **7.18*** | 95.42 | 95.06 | 6.64 | **6.52*** | 94.60 | 94.61 | 11.71 | **10.04*** | 94.58 | 94.58 |
| ToolAlpaca | 4 | **1.17** | 1.18 | 97.08 | 96.85 | **1.07** | 1.08 | 95.33 | 95.68 | 1.12 | **1.05*** | 95.68 | 95.44 |
| | 10 | 1.51 | **1.39*** | 95.21 | 95.56 | 1.25 | **1.20*** | 95.56 | 95.09 | 2.05 | **1.42*** | 95.56 | 94.51 |
| | 15 | 1.97 | **1.67*** | 96.50 | 96.03 | 1.68 | **1.54*** | 98.36* | 97.20 | 3.54 | **1.77*** | 96.14 | 95.21 |
| TruthfulQA | 4 | 3.34 | **2.69*** | 95.95* | 92.41 | 2.85 | **2.53*** | 96.71 | 96.71 | 2.74 | **1.88*** | 96.46 | 95.44 |
| | 10 | 7.06 | **6.41*** | 94.43 | 93.42 | 7.48 | **6.49*** | 98.48* | 95.70 | 7.52 | **5.64*** | 95.44 | 97.22 |
| | 15 | **10.61** | 10.62 | 94.68 | 94.68 | 10.72 | **10.30*** | 95.44 | 96.46 | 11.23 | **9.35*** | 95.44 | 96.46 |

Table 1: Average set sizes and coverage rates (in percentages) for conformal prediction sets on the MMLU, ToolAlpaca, and TruthfulQA datasets using `gemma-2-9b-it-SimPO` (Gemma-2), `Llama-3-8B-Instruct` (Llama-3) and `Phi-3-4k-mini-Instruct` (Phi-3), with a target coverage level of $95\%$. Bold numbers indicate smaller average set sizes. Asterisks on the larger of a pair of numbers indicate where the difference in average set size or coverage is statistically significant at the $0.05$ significance level.

| | | Llama-3 | | | Phi-3 | | | Gemma-2 | | |
|---|---|---|---|---|---|---|---|---|---|---|
| Model | # Opt. | Accuracy Before $(a_1)$ | Accuracy After $(a_1')$ | Gain $(a_1' - a_1)$ | Accuracy Before $(a_1)$ | Accuracy After $(a_1')$ | Gain $(a_1' - a_1)$ | Accuracy Before $(a_1)$ | Accuracy After $(a_1')$ | Gain $(a_1' - a_1)$ |
| MMLU | 4 | **64.02** | 63.83 | -0.19 | **70.27** | 69.08 | -1.19 | 67.62 | **67.70** | 0.07 |
| | 10 | 54.82 | **56.29** | 1.47* | 58.44 | **61.57** | 3.13* | 53.80 | **53.93** | 0.13 |
| | 15 | 51.99 | **54.11** | 2.11* | 53.48 | **58.09** | 4.62* | **50.78** | 50.58 | -0.20 |
| ToolAlpaca | 4 | 91.47 | **91.94** | 0.47 | **92.76** | 92.64 | -0.12 | **93.46** | 93.11 | -0.35 |
| | 10 | 85.16 | **88.67** | 3.50* | 87.50 | **90.89** | 3.39* | 87.73 | **89.60** | 1.87* |
| | 15 | 81.43 | **87.85** | 6.43* | 85.98 | **89.25** | 3.27* | 87.97 | **88.55** | 0.58 |
| TruthfulQA | 4 | 54.43 | **55.19** | 0.76 | 69.87 | **70.13** | 0.25 | 74.68 | **74.94** | 0.25 |
| | 10 | 39.24 | **40.76** | 1.52 | **55.70** | 54.43 | -1.27 | **56.46** | 56.20 | -0.25 |
| | 15 | 37.22 | **37.22** | 0.00 | **46.84** | 46.33 | -0.51 | 55.95 | **56.96** | 1.01 |

Table 2: [CROQ + logits]. Results on accuracy improvement with CROQ using logit scores. Here, $a_1$, and $a_1'$ refer to the accuracy before CROQ and after CROQ, respectively. A positive gain implies CROQ improved accuracy in that setting.

| | | Llama-3 | | | Phi-3 | | | Gemma-2 | | |
|---|---|---|---|---|---|---|---|---|---|---|
| Model | # Opt. | Accuracy Logits $(a_1')$ | Accuracy CP-OPT $(a_2')$ | Gain $(a_2' - a_1')$ | Accuracy Logits $(a_1')$ | Accuracy CP-OPT $(a_2')$ | Gain $(a_2' - a_1')$ | Accuracy Logits $(a_1')$ | Accuracy CP-OPT $(a_2')$ | Gain $(a_2' - a_1')$ |
| MMLU | 4 | **63.83** | 63.67 | -0.16 | 69.08 | **69.34** | 0.26 | 67.70 | **69.56** | 1.86* |
| | 10 | 56.29 | **57.11** | 0.82* | **61.57** | 61.05 | -0.52 | 53.93 | **57.93** | 4.00* |
| | 15 | 54.11 | **54.77** | 0.66* | 58.09 | **58.15** | 0.06* | 50.58 | **51.31** | 0.73 |
| ToolAlpaca | 4 | **91.94** | 91.82 | -0.12 | **92.64** | 92.52 | -0.12 | 93.11 | **93.57** | 0.46 |
| | 10 | 88.67 | **89.02** | 0.35* | 90.89 | **91.00** | 0.11* | 89.60 | **90.42** | 0.82* |
| | 15 | 87.85 | **88.67** | 0.82* | 89.25 | **89.95** | 0.70* | 88.55 | **89.37** | 0.82 |
| TruthfulQA | 4 | 55.19 | **55.44** | 0.25 | **70.13** | 69.87 | -0.26 | 74.94 | **76.96** | 2.02 |
| | 10 | 40.76 | **42.28** | 1.52 | 54.43 | **56.20** | 1.77 | 56.20 | **60.76** | 4.56* |
| | 15 | 37.22 | **37.47** | 0.25 | 46.33 | **51.39** | 5.06* | 56.96 | **57.72** | 0.76 |

Table 3: [CROQ + logits vs CROQ + CP-OPT]. Comparison of CP-OPT and logits on accuracy improvement with CROQ. Here, $a_1'$, and $a_2'$ are the final accuracies after CROQ using logits and CP-OPT respectively (as in Tables 2 and 5. The gain $a_2' - a_1'$ is the difference between these two, with values indicating more improvement in CROQ with CP-OPT scores.

options in the revised prompts. Based on Figure 1, we expect LLMs to be more likely to answer correctly when prompted with the revised question with fewer options. The results of CROQ with CP-OPT are summarized in Table 5, and in Table 3 we compare the accuracies after CROQ with logits and CP-OPT. In Table 3 we see that in 22 out of 27 settings, CROQ with CP-OPT results in higher accuracy (up to 4.56%) than CROQ with logits. Furthermore, the improvements in 12 out of these 22 settings are statistically significant. The drop in accuracy in the remaining 5 settings is statistically non-significant. Overall, the results show that CROQ with CP-OPT is generally better than with logits.

We provide additional experiments on the MMLU-Pro dataset in Appendix C.2 and the NL2SQL application in Appendix C.3.

## 5. Related Work

**Conformal Prediction for Uncertainty Quantification with LLMs.** Recently there has been growing interest in using conformal prediction to quantify and control uncertainty in LLM-related tasks. In the context of multi-choice question answering (MCQ), previous works have investigated a variety of conformal score functions, including (the softmax of) the LLM logits corresponding to the response options (Kumar et al., 2023; Ren et al., 2023) or functions thereof (Ye et al., 2024), confidence scores generated by the LLM itself, or "self-consistency" scores derived by repeated querying of the LLM (Su et al., 2024). We build on this work by aiming to learn a conformal score function that yields small conformal sets, rather than taking the score function as given.

In addition to the MCQ setting, there has been recent work utilizing conformal prediction in the context of open-ended response generation (Quach et al., 2024; Mohri & Hashimoto, 2024; Cherian et al., 2024). This setting differs in that there is not necessarily a unique correct response, so the notion of coverage must be redefined around *acceptability* or *factuality* rather than correctness. When factuality is the target, the goal is to calibrate a pruning procedure that removes a minimal number of claims from an LLM-generated open response, such that the remaining claims are all factual with high probability; that is, the goal is to retain as large a set as possible, rather than to generate a set with the smallest number of responses possible as in MCQ. Conformal prediction has also been used to capture token-level uncertainty (Deutschmann et al., 2024; Ravfogel et al.; Ulmer et al., 2024).

**Optimizing Conformal Prediction Procedures.** Several recent works have considered how to learn good conformal score functions from data, primarily in the context of supervised learning models (Bai et al., 2022; Stutz et al., 2022;

Yang & Kuchibhotla, 2024; Xie et al., 2024). With LLMs, Cherian et al. (2024) consider how to learn a good score function to achieve factuality guarantees; their optimization problem differs from ours due to the difference in setting as well as the addition of conditional coverage constraints (ensuring that coverage holds in different parts of the feature space). Kiyani et al. (2024) design a framework to minimize the size ("length," in their terminology) of conformal sets, which they apply to MCQ as well as to supervised learning problems. However, their framework is concerned with how to generate sets given a model and a conformity score, rather than how to learn a conformity score.

The works mentioned above all aim to produce small conformal sets that satisfy coverage guarantees. Among these, only Ren et al. (2023) consider how conformal sets may be used downstream, in their case to improve the efficiency and autonomy of robot behavior. To our knowledge, our work is the first to investigate whether conformal prediction can be used to increase the accuracy of LLMs on MCQ type tasks.

## 6. Conclusion and Future Work

In this work, we introduced Conformal Revision of Questions (CROQ), a principled approach to improve LLM accuracy in multiple-choice settings by leveraging conformal prediction (CP) to eliminate distractor answers while maintaining high coverage of the correct answer. To further boost CROQ's performance, we proposed CP-OPT, a framework for optimizing score functions to minimize prediction set sizes while preserving CP's coverage guarantees. Our results demonstrate that CROQ significantly enhances LLM's accuracy, and that CP-OPT further strengthens this effect by producing smaller, more reliable prediction sets than standard LLM logits. These findings highlight the potential of uncertainty-aware, test-time methods to improve LLM decision-making, providing a principled path for safer and more effective deployment of LLMs in critical applications.

Future works could explore multi-round CROQ, where answer options are pruned iteratively in multiple rounds, further improving accuracy while maintaining coverage. This requires developing efficient recalibration strategies and methods to prevent excessive coverage reduction across iterations. Additionally, a key challenge is adapting conformal score thresholds in settings with a variable number of response options. Techniques like quantile regression could help calibrate thresholds dynamically, ensuring robust performance across diverse decision-making scenarios.

## Impact Statement

This paper presents work whose goal is to advance the field of Machine Learning. There are many potential societal consequences of our work, none of which we feel must be specifically highlighted here.

## Acknowledgments

We are grateful to Sujay Bhatt, Alec Koppel, and Udari Madhushani Sehwag for fruitful discussions and feedback. We also appreciate the valuable inputs provided by the anonymous reviewers.

## Disclaimer

This paper was prepared for informational purposes by the Artificial Intelligence Research group of JPMorgan Chase & Co. and its affiliates ("JP Morgan") and is not a product of the Research Department of JP Morgan. JP Morgan makes no representation and warranty whatsoever and disclaims all liability, for the completeness, accuracy or reliability of the information contained herein. This document is not intended as investment research or investment advice, or a recommendation, offer or solicitation for the purchase or sale of any security, financial instrument, financial product or service, or to be used in any way for evaluating the merits of participating in any transaction, and shall not constitute a solicitation under any jurisdiction or to any person, if such solicitation under such jurisdiction or to such person would be unlawful.

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

## Supplementary Material

The supplementary material is organized as follows. In Appendix A.1 we provide details of LLM inference for MCQs. Appendix B provides proofs of the propositions in the paper. Additional experiments and results are given in Appendix C. First, in Appendix C.1 we discuss the trade-off between coverage (choice of $\alpha$) in conformal prediction and its effect on CROQ accuracy. In Appendix C.2 we provide results on the MMLU-Pro dataset, and in Appendix C.3 we provide details of experiments on a use-case in NL2SQL. Next, in Appendix C.4 we explore the effectiveness of conformal prediction with CP-OPT scores in deferral applications. The Appendices C.5,C.6, and C.7, contain more detailed results for the hypotheses discussed in the main paper. Appendix D provides details of the procedure used to compute statistical significance. In Appendix E we provide details of datasets and give samples of prompts before and after CROQ and LLM's answers. Finally, Appendix F lists the hyperparameters used for learning score function using CP-OPT.

## A. Methodology and Background Details

### A.1. Details on LLM inference in multi-choice question answering

We provide a formal description of the inference procedure described in the **LLM Inference** paragraph of Section 2.1.

The input prompt $x$ is a sequence of tokens $t_1, t_2, \ldots t_n$. We run the forward pass of the auto-regressive LLM (Touvron et al., 2023; Dubey et al., 2024; Abdin et al., 2024) on $x$ to produce a set of output logits:

$$\boldsymbol{l}_1, \boldsymbol{l}_2, \ldots, \boldsymbol{l}_n \leftarrow \texttt{LLM}\big(t_1, t_2, \ldots t_n\big) \tag{10}$$

Here, each logit $\boldsymbol{l}_j \in \mathbb{R}^{|V|}$ expresses the likelihood of the next token after $t_1, \ldots, t_j$, where $V$ is the universal set of tokens (aka the alphabet) for the given LLM and $|V|$ is its size. The last token's logits $\boldsymbol{l}_n$ are expected to have a high value for the correct answer key. We extract the logit vector $\bar{\boldsymbol{l}} \in \mathbb{R}^m$ corresponding to the option keys as follows:

$$\bar{\boldsymbol{l}} := \big[\, \boldsymbol{l}_n[Y_1],\, \boldsymbol{l}_n[Y_2],\, \ldots, \boldsymbol{l}_n[Y_m] \,\big], \tag{11}$$

where $\boldsymbol{l}_n[Y_j]$ denotes the logit value corresponding to the token $Y_j$ in the last token's logits $\boldsymbol{l}_n$. The logits $\bar{\boldsymbol{l}}$ are converted to softmax scores $s(x)$. The softmax score of point $x$ and option key $y$ is denoted by $s(x, y)$ and the predicted answer key $\hat{y}$ corresponds to the maximum softmax value:

$$s(x) := \texttt{softmax}(\bar{\boldsymbol{l}}), \qquad s(x, y) := s(x)[y], \qquad \hat{y} := \underset{y \in \{Y_1, \ldots Y_m\}}{\arg\max} \; s(x, y) \tag{12}$$

## B. Proofs

### B.1. Proof of Proposition 2.1

The proof is nearly identical to the proof of Theorem D.1 in Angelopoulos & Bates (2022), with intuitive modifications due to the fact that we use conformal scores (where higher scores indicate more plausible candidate answers) rather than nonconformity scores (where higher scores indicate less plausible candidates). We include it here for completeness.

*Proof.* Given a fixed conformal score function $g$, let $g_i = g(x_i, y_i^\star)$ for $i = 1, \ldots, n_{\text{cal}}$ denote the conformal scores on the calibration dataset, and denote the new sample from the same distribution (the "test sample") by $(\tilde{x}, \tilde{y}^\star)$. Without loss of generality, assume the scores are sorted, with ties resolved at random, so that $g_1 \leq \ldots \leq g_{n_{\text{cal}}}$. If $\alpha < 1/(n_{\text{cal}} + 1)$, then the conformal threshold is given by $\hat{\tau}_\alpha = -\infty$, in which case the conformal sets are equal to the output space $\mathcal{Y}_m$, and coverage is guaranteed. In case $\alpha \geq 1/(n_{\text{cal}} + 1)$, we have equality of the following two events:

$$\{\tilde{y}^\star \in \mathcal{C}(\tilde{x}; g, \hat{\tau}_\alpha)\} = \big\{g(\tilde{x}, \tilde{y}^\star) \geq g_{\lfloor (n_{\text{cal}}+1)\alpha \rfloor}\big\}. \tag{13}$$

Because all the samples are iid, we have for any integer $k$ that

$$\mathbb{P}\left(g(\tilde{x}, \tilde{y}^\star) \geq g_k\right) = \frac{n_{\text{cal}} - k + 1}{n_{\text{cal}} + 1} \tag{14}$$

where this probability is marginal over the randomness in the calibration dataset and the test sample $(\tilde{x}, \tilde{y}^\star)$. We therefore have

$$\mathbb{P}\left(g(\tilde{x}, \tilde{y}^\star) \geq g_{\lfloor (n_{\text{cal}}+1)\alpha \rfloor}\right) = \frac{n_{\text{cal}} - \lfloor (n_{\text{cal}}+1)\alpha \rfloor + 1}{n_{\text{cal}}+1} = 1 - \frac{\lfloor (n_{\text{cal}}+1)\alpha \rfloor}{n_{\text{cal}}+1} \geq 1 - \alpha \qquad (15)$$

which yields the desired coverage result. □

Note that the assumption that the samples are iid can be weakened to the assumption that the calibration samples and test sample are exchangeable, since property (14) still holds in this case.

This proof can be generalized to upper bound the coverage probability, which guarantees that the conformal sets are not overly conservative. See for example Lei et al. (2018, Thm. 2.2) and Tibshirani et al. (2019, Thm. 1).

### B.2. Proof of Proposition 3.2

The first claim follows from the law of total probability:

$$\mathbb{P}(\hat{y} = y^\star) = \sum_{k=1}^{m} \mathbb{P}\left(\hat{y} = y^\star \mid \nu(x) = k, y^\star \in C(x; g, \hat{\tau}_\alpha)\right) \mathbb{P}\left(y^\star \in C(x; g, \hat{\tau}_\alpha) \mid \nu(x) = k\right) \mathbb{P}\left(\nu(x) = k\right)$$

$$= \sum_{k=1}^{m} f_{\text{post}}(k)\rho_k r_k.$$

The second claim follows by writing the change in accuracy due to CROQ as

$$\Delta(f_{\text{post}}, \alpha, a) = \sum_{k=1}^{m} r_k \rho_k f_{\text{post}}(k) - a = \sum_{k=1}^{m} \left(r_k \rho_k f_{\text{post}}(k) - \frac{1}{m}a\right)$$

and noting that if each of the $m$ terms in the rightmost sum is positive, then the overall sum is positive. The final claim follows from the equivalence with the fractional knapsack problem (Cormen et al., 2009, Ch. 16.2). In terms of the knapsack problem, here our items are the groups of questions corresponding to sizes $1, \ldots, m$. For each item $k$, the value function is the accuracy $f(k)$, and we want to pick as much as possible from each group. The quantity $r_k \rho_k$ denotes the "effective fraction" of the value picked up. Thus, the sequence $r_k \rho_k$ such that $r_1 \rho_1 \geq r_2 \rho_2 \geq \ldots \geq r_m \rho_m$ subject to the constraints $\sum_{k=1}^{m} r_k \leq 1$ and $1 - \alpha \leq \sum_{k=1}^{m} r_k \rho_k \leq 1$ will maximize the accuracy after CROQ, in other words maximize the gain $\Delta(f_{\text{post}}, \alpha, a)$.

We note that in practice, $r_k, \rho_k$, and $f_{\text{post}}(k)$ will all depend on one another, since $f_{\text{post}}(k)$ will depend on the distribution of set sizes across different types of questions. This analysis is intended to provide insight and suggest directions for more in-depth future analyses.

## C. Additional Experiments and Results

This appendix contains additional results and details not included in the main paper due to length constraints.

### C.1. Trade-off between coverage and accuracy

The choice of $\alpha$ controls the coverage level in conformal prediction. A small $\alpha$ implies high coverage, meaning the prediction sets contain the true options with high probability but potentially have large sizes. Thus, choosing a very small $\alpha$ will likely not eliminate a sufficient number of options to see any noticeable improvement with CROQ. On the other hand, choosing a large $\alpha$ will eliminate the true option from the set for a large portion of the questions, which will result in low accuracy from CROQ. To study these trade-offs, we run CROQ with different values of $\alpha$. The accuracy before and after CROQ for a range of $\alpha$ values are shown in Figure 5 and Figure 4 for the Llama-3 and Phi-3 models, respectively. The results are as expected given the observations above: using an overly conservative (small) $\alpha$ does not give much improvement; as we increase $\alpha$, the accuracy also increases up to a point, after which it starts to come down. This suggests that to optimize accuracy, a practitioner can tune $\alpha$ for their chosen score function and setting.

|  | Accuracy | Avg. Set Size | Coverage | LLM Cost |
|---|---|---|---|---|
| **Approach 1** | 32.0% | 7.270 | 100% | $7.10 |
| **Approach 2** | 29.5% | 6.405 | 88% | $6.63 |
| **Approach 3** | 32.5% | **2.685** | 92% | **$3.89** |

Table 4: Results with different approaches on the table selection step in the NL2SQL task.

## C.2. Evaluation on MMLU-Pro

We evaluated CROQ on the MMLU-Pro (Wang et al.) dataset with questions having 10 options. We observe that the baseline accuracy with the Phi-3 model is 36.4%, and we get a 3% relative improvement in accuracy with CROQ – a significant improvement on a 10-option dataset, particularly given that MMLU-pro contains much harder questions.

## C.3. Application to an agentic workflow on NL2SQL

For an application in an agentic workflow, we consider the Natural Language Question to SQL (NL2SQL) task, where an LLM-based agent generates a SQL query for a user's natural language question. A component of the standard agentic workflow in this task is to first predict the relevant tables whose schema should be included in the context of the LLM, which generates the SQL query. This step is critical to decrease cost and, in some cases, is necessary when the full database schema would exceed the LLM's context limit.

We consider the BIRD dataset (Li et al., 2023) - a large benchmark that contains 12,751 NLQ-SQL pairs across 95 databases. We filter out databases with 20 tables or more (to avoid context limit errors) and remove the retail_world databases due to inconsistent table naming. We considered the following settings:

**Approach 1** - Include all table schemas in the LLM prompt.

**Approach 2** - Include all table schemas for tables whose cosine similarity score is greater than a particular threshold, up to a maximum of 10 tables. The cosine similarity is taken between the embeddings of the natural language question and the table name using the OpenAI text-embedding-ada-002 model. Coverage is defined to include all tables used in the annotated ground-truth SQL query. Coverage was approximately 90%, although this was not explicitly controlled.

**Approach 3** - Include tables selected using conformal prediction (CP) on CP-OPT scores. This is equivalent to the CROQ procedure, where the scores for CP are obtained from a source other than LLM. More specifically, we learn CP-OPT scores using embeddings of natural language questions and table names.

We used 3412 NLQ-SQL pairs for training in approach 3, and validated on 3411 examples in approaches 2 and 3. We then tested the 3 approaches on 200 NLQ-SQL pairs. We use GPT4-0613 as the LLM for SQL query generation, and report the execution accuracy, average set size, and total token cost. The results in all three settings are summarized in Table 4. Here, the set size means the number of tables whose schema will be included in the LLM context. Thus, a lower avg. set size means fewer tables (and hence fewer tokens) in the LLM context. In the results, we see a significant reduction in the avg. set size in approach 3 while maintaining high coverage (92%). This results in a substantial reduction in the number of tokens in the LLM context, leading to a 45% decrease in LLM cost, all while achieving slightly higher accuracy in comparison to approach 1.

## C.4. Using conformal prediction for deferral

***Smaller prediction sets imply fewer deferrals in human-in-the-loop or model cascade systems.*** We consider a deferral procedure in which a set size cutoff is selected, and the LLM answer is only retained if the set size is at or below that cutoff. For all larger sets, the question is passed to a human (or a more powerful but costly model) who can answer the question correctly. Smaller sets from CP are desirable for this procedure to be effective. We evaluate this procedure with logit and CP-OPT scores in two settings and show the results in Figure 3. As expected, lower set size cutoffs result in higher accuracy. As the set size cutoff increases, the accuracy approaches the LLM's marginal accuracy, while the number of deferrals (i.e., the cost of obtaining the answer from a human or more expensive model) decreases. In the top row of the figure, the differences in the set sizes between logit and CP-OPT scores are not large enough to see a meaningful difference in this

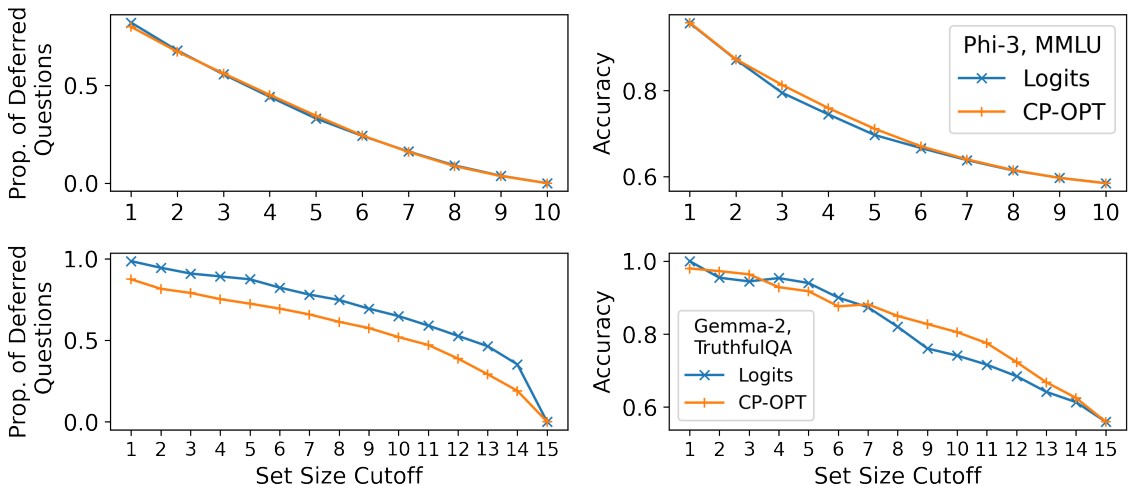

Figure 3: Proportion of questions deferred to a human when conformal prediction set sizes exceed a certain cutoff (left), and the corresponding LLM accuracy for questions (without revision) retained by the LLM as a function of cutoff threshold (right). In the top row (MMLU, 10 options, `Phi-3-4k-mini-Instruct`), the difference in deferral and accuracy is negligible, whereas in the bottom row (TruthfulQA, 15 options, `gemma-2-9b-it-SimPO`), CP-OPT defers fewer questions to the human while providing similar or improved accuracy for questions retained.

procedure. However, in the bottom row corresponding to the Gemma-2 model and TruthfulQA dataset with 15 options, we see CP-OPT scores lead to fewer deferrals in comparison to logits. Model cascades (Dohan et al., 2022; Gupta et al., 2024) and deferrals to human-in-the-loop (Tailor et al., 2024; Vishwakarma et al., 2024) and more broadly selective prediction (El-Yaniv & Wiener, 2010; Fisch et al., 2022; Vishwakarma et al., 2023) are useful frameworks for model deployment while ensuring safety, high accuracy, and balancing the costs. Our experiments show the promise of CP with logit and CP-OPT scores in this task and suggest it would be fruitful to explore this design space with CP.

Figure 4 shows accuracy after the CROQ procedure as a function of $\alpha$ for Phi-3. The results are qualitatively similar to the results for Llama-3 in the main text (Section 4.2).

All remaining results are organized by dataset. Tables for the CROQ results, which illustrate accuracy changes conditional on set size are based on a confidence level of $95\%$ (equivalently, an $\alpha$ level of 0.05). Note that with the ToolAlpaca dataset, not all possible set sizes occur, in which case we omit the corresponding columns. For example, with 10 response options, only sets of size 8 and smaller occur.

Asterisks in the tables indicate where the difference in overall accuracy from Before to After, i.e,. from baseline to after the CROQ procedure, is statistically significant at the $\alpha = 0.05$ level. (In some tables, like Table 9, none of the changes are significant.) See Appendix D for details on how statistical significance was calculated.

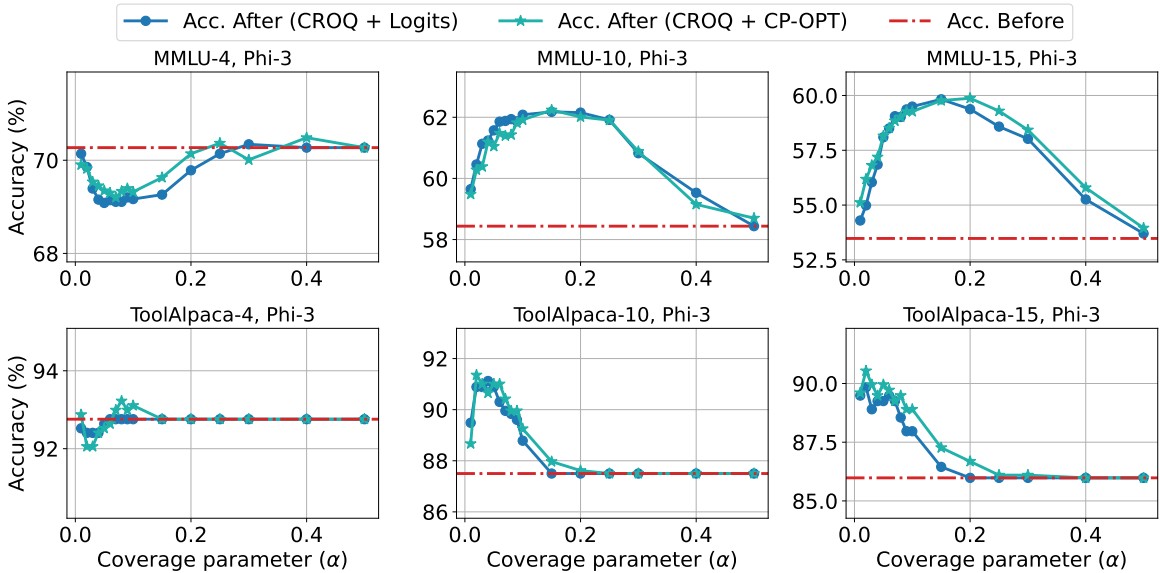

Figure 4: Accuracy on revised questions on the MMLU and ToolAlpaca datasets while varying miscoverage parameter $\alpha$ for `Phi-3-4k-mini-Instruct` (Phi-3) model and both scores. Smaller values of $\alpha$ correspond to high levels of coverage. When coverage is too large, few or no answers are eliminated, and the LLM is prompted with the same question. When coverage is low, a larger portion of answer sets no longer contain the true answer and the benefits of revision are diminished.

## C.5. MMLU

Results for the experiments on the MMLU dataset are given in Tables 9 and 10,Tables 6 to 8 and Figures 6 to 8.

## C.6. TruthfulQA

Results for the experiments on the TruthfulQA dataset are given in Tables 11 to 15 and Figures 13 and 14.

## C.7. ToolAlpaca

Results for experiments on the ToolAlpaca dataset are given in Tables 16 to 20 and Figures 10 and 11.

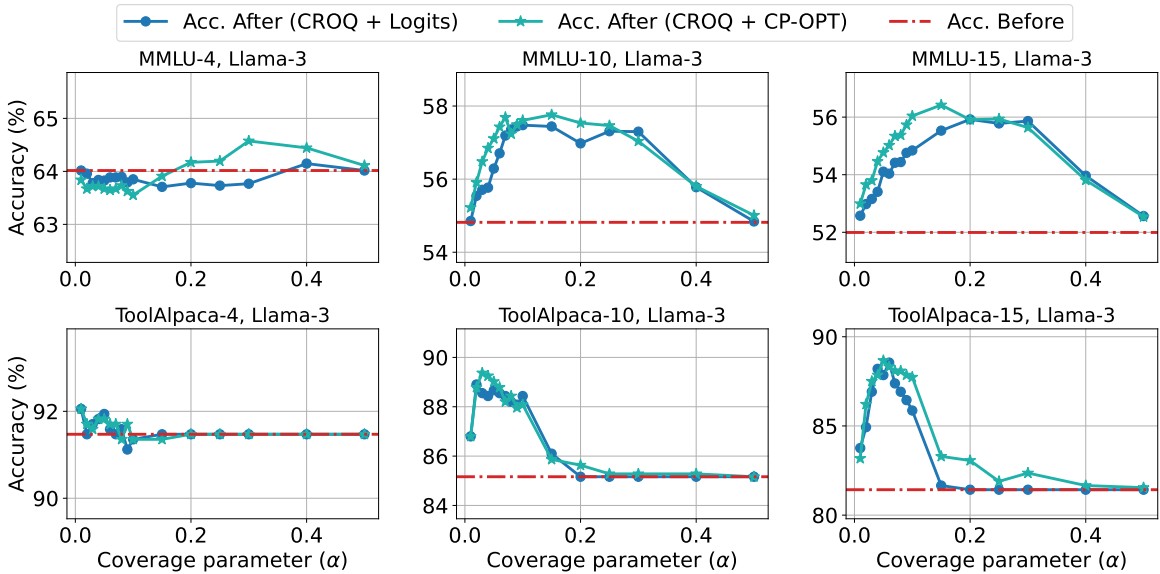

Figure 5: Accuracy on revised questions on the MMLU and ToolAlpaca datasets while varying miscoverage parameter $\alpha$ for `Llama-3-8B-Instruct` (Llama-3) model and both scores. Smaller values of $\alpha$ correspond to high levels of coverage. When coverage is too large, few or no answers are eliminated, and the LLM is prompted with the same question. When coverage is low, a larger portion of answer sets no longer contain the true answer or produce empty prediction sets thus resulting in diminished benefits of revision.

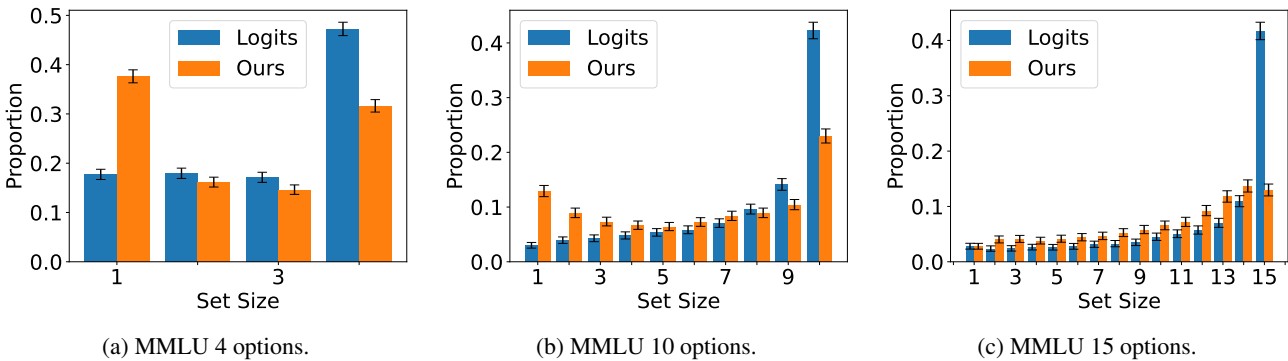

(a) MMLU 4 options.

(b) MMLU 10 options.

(c) MMLU 15 options.

Figure 6: Distributions of sizes of sets obtained from CP-OPT and logit scores on MMLU dataset and Gemma-2 model.

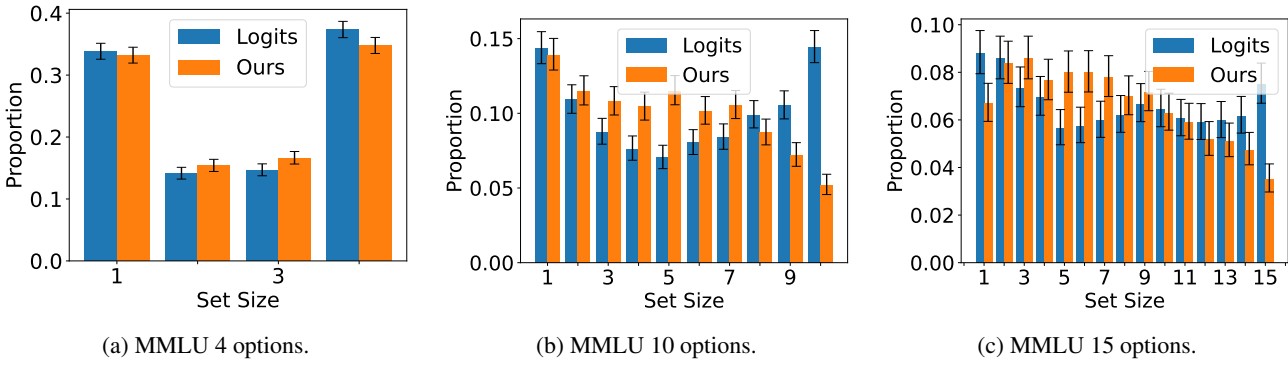

(a) MMLU 4 options.

(b) MMLU 10 options.

(c) MMLU 15 options.

Figure 7: Distributions of sizes of sets obtained from CP-OPT and logit scores on MMLU dataset and Llama-3 model.

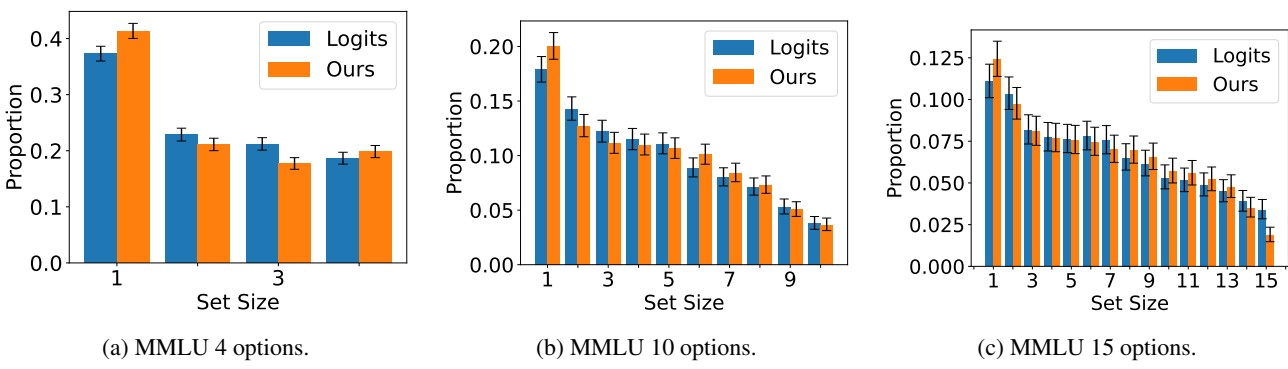

(a) MMLU 4 options.

(b) MMLU 10 options.

(c) MMLU 15 options.

Figure 8: Distributions of sizes of sets obtained from CP-OPT and logit scores on MMLU dataset and Phi-3 model setting.

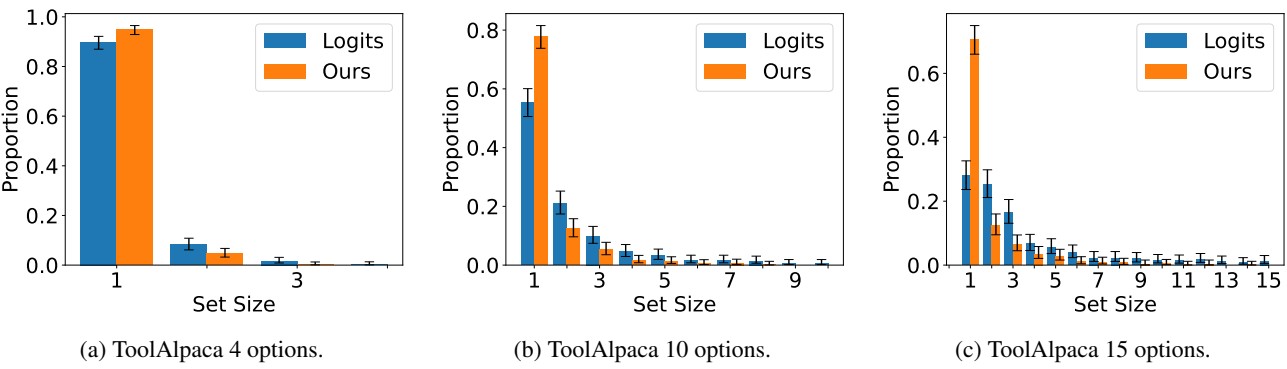

(a) ToolAlpaca 4 options.

(b) ToolAlpaca 10 options.

(c) ToolAlpaca 15 options.

Figure 9: Distributions of sizes of sets obtained from CP-OPT and logit scores on ToolAlpaca dataset and Gemma-2 model.

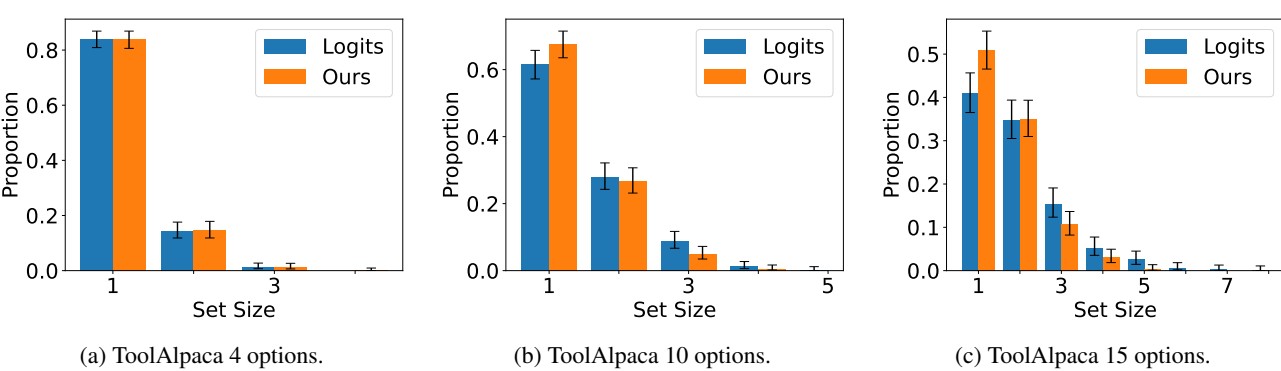

(a) ToolAlpaca 4 options.

(b) ToolAlpaca 10 options.

(c) ToolAlpaca 15 options.

Figure 10: Distributions of sizes of sets obtained from CP-OPT and logit scores on ToolAlpaca dataset and Llama-3 model.

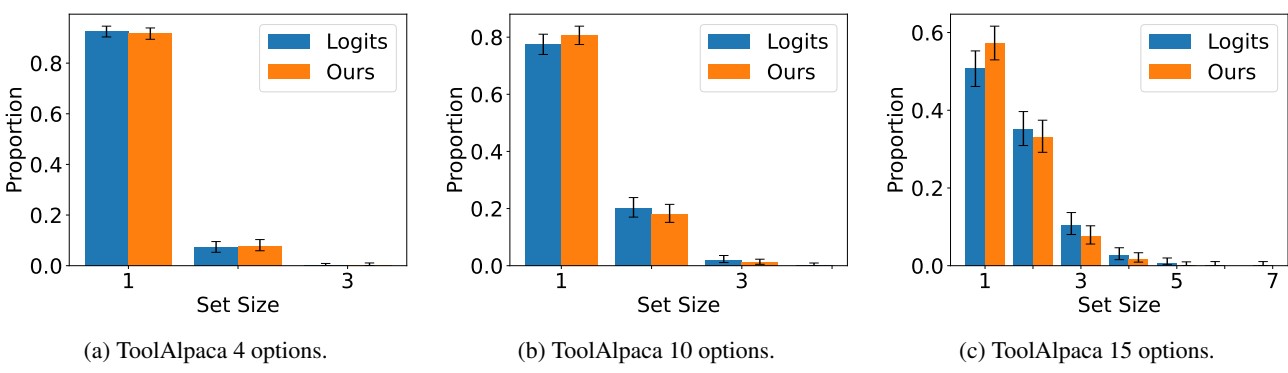

(a) ToolAlpaca 4 options.

(b) ToolAlpaca 10 options.

(c) ToolAlpaca 15 options.

Figure 11: Distributions of sizes of sets obtained from CP-OPT and logit scores on ToolAlpaca dataset and Phi-3 model.

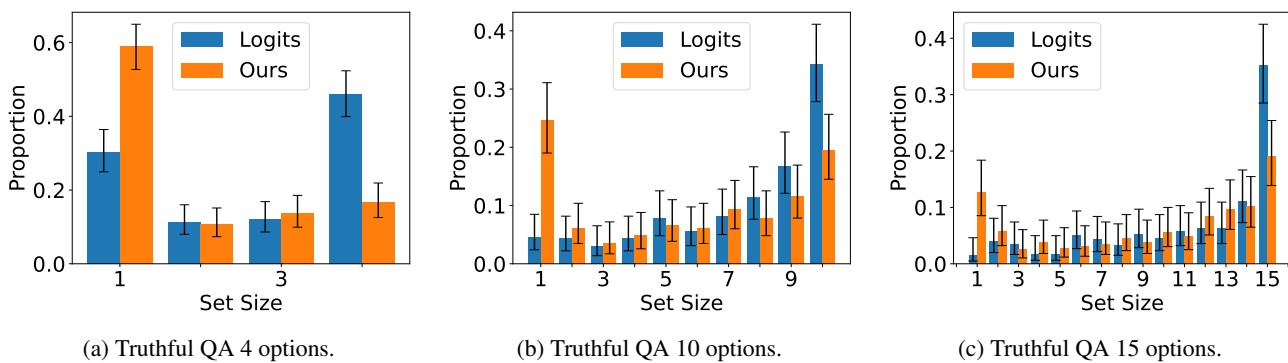

(a) Truthful QA 4 options.

(b) Truthful QA 10 options.

(c) Truthful QA 15 options.

Figure 12: Distributions of sizes of sets obtained from CP-OPT and logit scores on Truthful QA dataset and Gemma-2.

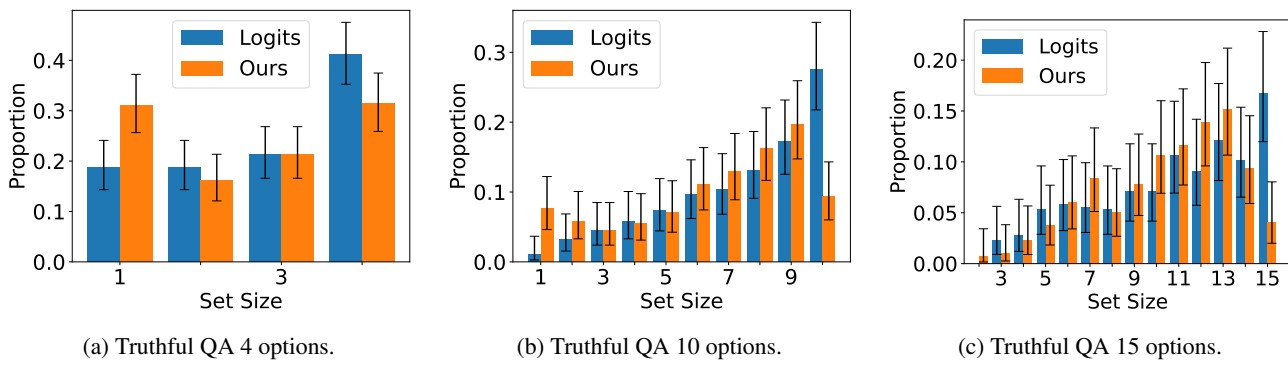

(a) Truthful QA 4 options.

(b) Truthful QA 10 options.

(c) Truthful QA 15 options.

Figure 13: Distributions of sizes of sets obtained from CP-OPT and logit scores on Truthful QA dataset and Phi-3 model.

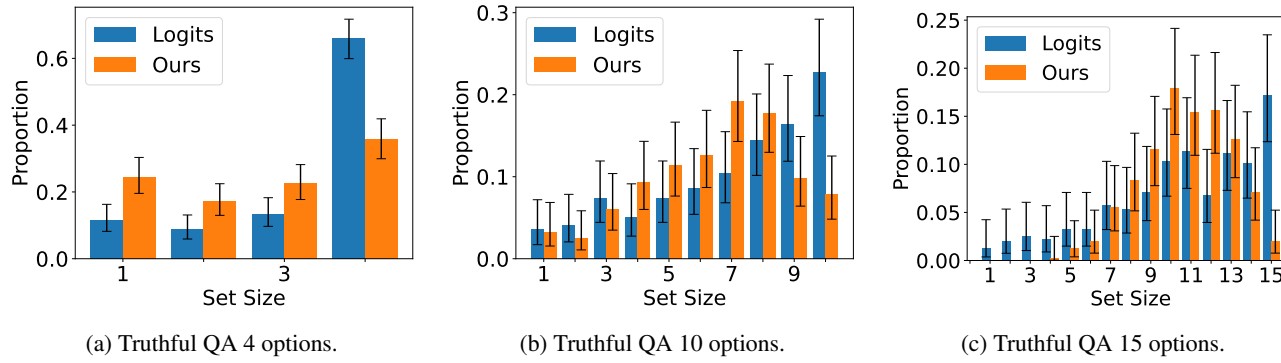

(a) Truthful QA 4 options.  (b) Truthful QA 10 options.  (c) Truthful QA 15 options.

Figure 14: Distributions of sizes of sets obtained from CP-OPT and logit scores on Truthful QA dataset and Llama-3 model.

| Model | # Opt. | LLama-3 | | | Phi-3 | | | Gemma-2 | | |
|-------|--------|---------|---------|---------|---------|---------|---------|---------|---------|---------|
| | | Accuracy Before $(a_2)$ | Accuracy After $(a_2')$ | Gain $(a_2' - a_2)$ | Accuracy Before $(a_2)$ | Accuracy After $(a_2')$ | Gain $(a_2' - a_2)$ | Accuracy Before $(a_2)$ | Accuracy After $(a_2')$ | Gain $(a_2' - a_2)$ |
| **MMLU** | 4 | **64.02** | 63.67 | -0.34 | **70.27** | 69.34 | -0.93 | 68.36 | **69.56** | 1.20* |
| | 10 | 54.82 | **57.11** | 2.29* | 58.44 | **61.05** | 2.61* | 53.99 | **57.93** | 3.94* |
| | 15 | 51.99 | **54.77** | 2.78* | 53.48 | **58.15** | 4.68* | 50.78 | **51.31** | 0.52 |
| **ToolAlpaca** | 4 | 91.47 | **91.82** | 0.35 | **92.64** | 92.52 | -0.12 | 93.46 | **93.57** | 0.12 |
| | 10 | 85.16 | **89.02** | 3.86* | 87.62 | **91.00** | 3.39* | 88.08 | **90.42** | 2.34* |
| | 15 | 81.43 | **88.67** | 7.24* | 85.98 | **89.95** | 3.97* | 88.08 | **89.37** | 1.29 |
| **TruthfulQA** | 4 | 54.43 | **55.44** | 1.01 | 69.87 | **69.87** | 0.00 | 74.94 | **76.96** | 2.03 |
| | 10 | 39.24 | **42.28** | 3.04 | 55.70 | **56.20** | 0.51 | 56.46 | **60.76** | 4.30* |
| | 15 | 37.22 | **37.47** | 0.25 | 46.84 | **51.39** | 4.56* | 55.95 | **57.72** | 1.77 |

Table 5: [CROQ + CP-OPT]. Results on accuracy improvement with CROQ using CP-OPT scores. Here $a_2$, and $a_2'$ refer to the accuracy before CROQ and after CROQ respectively. A higher gain in a setting suggests CROQ improved accuracy in that setting.

| Score | Set Size | 1 | 2 | 3 | 4 | 5 | 6 | 7 | 8 | 9 | 10 | 11 | 12 | 13 | 14 | 15 | Overall |
|-------|----------|---|---|---|---|---|---|---|---|---|----|----|----|----|----|----|---------|
| Logits | Coverage | 82.40 | 69.04 | 80.00 | 83.56 | 81.11 | 87.45 | 86.31 | 88.60 | 90.75 | 90.45 | 94.80 | 93.75 | 98.30 | 98.15 | 100 | 94.58 |
| | Fraction | 2.77 | 2.34 | 2.37 | 2.60 | 2.58 | 2.74 | 3.12 | 3.23 | 3.47 | 4.47 | 5.02 | 5.70 | 6.99 | 10.91 | 41.70 | 100 |
| | Acc. Before | 82.40 | 62.44 | 62.00 | 65.30 | 60.37 | **61.47** | **61.98** | 59.19 | 55.82 | **62.6** | **57.92** | 51.25 | 57.89 | **50.38** | 40.01 | **50.78** |
| | Acc. After | 82.40 | 65.48 | 68.50 | 65.75 | 63.13 | 58.87 | 60.08 | 57.72 | **56.85** | 58.89 | 55.08 | **51.88** | **58.06** | 49.40 | **40.01** | 50.58 |
| Ours | Coverage | 93.10 | 94.05 | 89.83 | 89.94 | 89.34 | 90.54 | 89.74 | 90.23 | 92.40 | 94.73 | 94.70 | 94.46 | 96.77 | 97.74 | 100 | 94.58 |
| | Fraction | 2.75 | 3.99 | 4.08 | 3.77 | 4.12 | 4.39 | 4.63 | 5.22 | 5.78 | 6.53 | 7.17 | 9.21 | 11.76 | 13.66 | 12.94 | 100 |
| | Acc. Before | 93.10 | 88.10 | 82.56 | 79.56 | **75.79** | **73.24** | **64.62** | **56.82** | 56.26 | 52.73 | 45.20 | 42.53 | 36.63 | 33.10 | 25.96 | 50.78 |
| | Acc. After | 93.10 | 89.58 | 82.56 | 80.82 | 73.78 | 70.81 | 60.26 | 56.14 | **57.49** | **53.27** | **46.69** | **43.94** | **40.06** | **33.80** | 25.96 | **51.31** |

Table 6: Results for CROQ on the MMLU dataset with 15 response options and Gemma-2 model.

| Score | Set Size | 1 | 2 | 3 | 4 | 5 | 6 | 7 | 8 | 9 | 10 | 11 | 12 | 13 | 14 | 15 | Overall |
|-------|----------|---|---|---|---|---|---|---|---|---|----|----|----|----|----|----|---------|
| Logits | Coverage | 95.82 | 91.56 | 89.98 | 93.19 | 94.54 | 94.63 | 94.44 | 95.60 | 96.09 | 96.88 | 97.06 | 96.77 | 98.21 | 98.08 | 100 | 95.42 |
| | Fraction | 8.81 | 8.58 | 7.35 | 6.97 | 5.65 | 5.74 | 5.98 | 6.21 | 6.68 | 6.46 | 6.05 | 5.89 | 5.97 | 6.17 | 7.50 | 100 |
| | Acc. Before | 95.82 | 82.16 | 72.37 | 66.95 | 55.88 | 50.62 | 50.20 | **46.08** | 40.14 | 37.32 | 34.90 | 34.68 | 30.62 | **27.88** | 24.05 | 51.99 |
| | Acc. After | **95.82** | **83.82** | **76.09** | **71.55** | **63.66** | **53.93** | **51.39** | 45.32 | **43.69** | **40.99** | **36.47** | **35.08** | **33.00** | 27.69 | **24.05** | 54.11* |
| Ours | Coverage | 94.15 | 94.62 | 91.29 | 91.63 | 93.31 | 93.18 | 94.52 | 96.43 | 97.02 | 96.42 | 97.59 | 96.56 | 97.91 | 98.25 | 100 | 95.06 |
| | Fraction | 6.69 | 8.38 | 8.58 | 7.65 | 7.99 | 8.00 | 7.80 | 6.99 | 7.17 | 6.30 | 5.90 | 5.17 | 5.12 | 4.75 | 3.51 | 100 |
| | Acc. Before | 94.15 | 87.54 | 73.58 | 65.58 | 55.57 | 51.78 | 45.81 | 46.86 | 39.90 | 31.83 | 33.00 | 28.67 | **31.32** | 21.25 | 19.59 | 51.99 |
| | Acc. After | **94.15** | **89.24** | **75.80** | **70.39** | **63.74** | **54.60** | **50.53** | **47.54** | **42.38** | **35.03** | **34.21** | **33.26** | 29.93 | **24.75** | 19.59 | 54.77* |

Table 7: Results for CROQ on the MMLU dataset with 15 response options and Llama-3 model.

| Score | Set Size | 1 | 2 | 3 | 4 | 5 | 6 | 7 | 8 | 9 | 10 | 11 | 12 | 13 | 14 | 15 | Overall |
|---|---|---|---|---|---|---|---|---|---|---|---|---|---|---|---|---|---|
| | **Coverage** | 96.03 | 92.77 | 93.46 | 91.71 | 93.93 | 93.61 | 93.55 | 93.81 | 94.79 | 96.65 | 95.38 | 96.83 | 95.77 | 97.25 | 100 | 94.60 |
| Logits | **Fraction** | 11.07 | 10.34 | 8.17 | 7.73 | 7.62 | 7.80 | 7.55 | 6.52 | 6.15 | 5.32 | 5.14 | 4.87 | 4.49 | 3.88 | 3.38 | 100 |
| | **Acc. Before** | 96.03 | 80.48 | 69.62 | 59.14 | 53.12 | 46.27 | 42.61 | 42.08 | 37.84 | 39.51 | 36.72 | 34.15 | 23.02 | 23.55 | 21.75 | 53.48 |
| | **Acc. After** | **96.03** | **84.85** | **76.60** | **66.97** | **63.86** | **53.42** | **51.10** | **44.44** | **42.86** | **42.19** | **39.26** | **36.34** | **25.13** | **24.46** | **21.75** | **58.09*** |
| | **Coverage** | 95.79 | 92.20 | 93.83 | 91.19 | 94.19 | 93.79 | 95.93 | 94.54 | 94.57 | 96.04 | 93.82 | 96.80 | 96.26 | 97.29 | 100 | 94.61 |
| Ours | **Fraction** | 12.40 | 9.73 | 8.08 | 7.68 | 7.56 | 7.45 | 7.00 | 6.95 | 6.55 | 5.70 | 5.57 | 5.20 | 4.76 | 3.50 | 1.86 | 100 |
| | **Acc. Before** | 95.79 | 80.24 | 73.86 | 60.28 | 51.33 | 49.68 | 43.90 | 41.47 | 36.41 | 31.46 | 29.42 | 29.00 | 25.69 | 21.69 | 18.47 | 53.48 |
| | **Acc. After** | **95.79** | **83.66** | **78.12** | **69.86** | **62.64** | **54.62** | **52.03** | **47.95** | **39.67** | **38.96** | **32.41** | **31.28** | **27.18** | **22.37** | **18.47** | **58.15*** |

Table 8: Results for CROQ on the MMLU dataset with 15 response options and Phi-3 model.

| Model | Score | Set Size | 1 | 2 | 3 | 4 | Overall |
|---|---|---|---|---|---|---|---|
| Gemma-2 | Logits | **Coverage** | 89.34 | 89.94 | 93.27 | 100 | 95.16 |
| | | **Fraction** | 17.71 | 17.93 | 17.11 | 47.25 | 100 |
| | | **Acc. Before** | 89.34 | 79.42 | **68.24** | 54.79 | 67.62 |
| | | **Acc. After** | **89.34** | **79.95** | 68.10 | **54.79** | **67.70** |
| | Ours | **Coverage** | 91.67 | 89.93 | 93.10 | 100 | 94.23 |
| | | **Fraction** | 37.62 | 16.14 | 14.61 | 31.63 | 100 |
| | | **Acc. Before** | 91.67 | 72.50 | 57.27 | 43.64 | 68.36 |
| | | **Acc. After** | **91.67** | **75.88** | **61.74** | **43.64** | **69.56*** |
| Llama-3 | Logits | **Coverage** | 93.55 | 92.78 | 92.89 | 100 | 95.75 |
| | | **Fraction** | 33.84 | 14.13 | 14.68 | 37.35 | 100 |
| | | **Acc. Before** | 93.55 | 70.19 | **49.88** | 40.48 | **64.02** |
| | | **Acc. After** | **93.55** | **70.70** | 48.10 | **40.48** | 63.83 |
| | Ours | **Coverage** | 93.71 | 91.83 | 93.50 | 100 | 95.57 |
| | | **Fraction** | 33.21 | 15.39 | 16.63 | 34.77 | 100 |
| | | **Acc. Before** | 93.71 | **71.16** | **52.46** | 38.02 | **64.02** |
| | | **Acc. After** | **93.71** | 70.01 | 51.46 | **38.02** | 63.67 |
| Phi-3 | Logits | **Coverage** | 94.75 | 91.48 | 93.17 | 100 | 94.65 |
| | | **Fraction** | 37.30 | 22.86 | 21.20 | 18.64 | 100 |
| | | **Acc. Before** | 94.75 | **70.25** | **52.69** | 41.31 | **70.27** |
| | | **Acc. After** | **94.75** | 66.93 | 50.67 | **41.31** | 69.08 |
| | Ours | **Coverage** | 93.63 | 90.61 | 94.17 | 100 | 94.35 |
| | | **Fraction** | 41.36 | 21.10 | 17.71 | 19.83 | 100 |
| | | **Acc. Before** | 93.63 | **67.38** | **52.82** | 40.22 | **70.27** |
| | | **Acc. After** | **93.63** | 64.57 | 50.94 | **40.22** | 69.34 |

Table 9: Results for CROQ on the MMLU dataset with 4 response options.

| Model | Score | Set Size | 1 | 2 | 3 | 4 | 5 | 6 | 7 | 8 | 9 | 10 | Overall |
|---|---|---|---|---|---|---|---|---|---|---|---|---|---|
| Gemma-2 | Logits | Coverage | 78.80 | 79.03 | 84.92 | 88.56 | 85.30 | 92.64 | 94.09 | 96.41 | 97.22 | 100 | 95.00 |
| | | Fraction | 2.97 | 3.90 | 4.25 | 4.77 | 5.33 | 5.80 | 7.03 | 9.59 | 14.10 | 42.26 | 100 |
| | | Acc. Before | 78.80 | 73.86 | 74.02 | 68.41 | 62.36 | **67.69** | 61.49 | 58.42 | **51.94** | 41.81 | 53.80 |
| | | Acc. After | 78.80 | **76.90** | **75.98** | **72.39** | **62.36** | 66.67 | 60.14 | 57.67 | 51.68 | **41.81** | 53.93 |
| | Ours | Coverage | 90.79 | 92.27 | 88.31 | 90.54 | 89.80 | 91.30 | 92.05 | 95.60 | 97.49 | 100 | 94.04 |
| | | Fraction | 12.89 | 8.90 | 7.31 | 6.65 | 6.40 | 7.23 | 8.36 | 8.90 | 10.41 | 22.96 | 100 |
| | | Acc. Before | 90.79 | 84.93 | 69.97 | 66.07 | 54.17 | 48.60 | 42.76 | 40.00 | 37.74 | 31.27 | 53.99 |
| | | Acc. After | **90.79** | **89.20** | **79.87** | **75.00** | **64.01** | **55.34** | **47.02** | **45.33** | **40.59** | 31.27 | 57.93* |
| Llama-3 | Logits | Coverage | 94.55 | 91.96 | 91.73 | 94.09 | 94.94 | 97.19 | 97.32 | 97.72 | 99.32 | 100 | 96.06 |
| | | Fraction | 14.36 | 10.92 | 8.76 | 7.63 | 7.04 | 8.03 | 8.40 | 9.90 | 10.53 | 14.43 | 100 |
| | | Acc. Before | 94.55 | **80.43** | 65.99 | 57.54 | 51.43 | 47.56 | 37.71 | 35.13 | 34.84 | 31.41 | 54.82 |
| | | Acc. After | 94.55 | 80.33 | **69.51** | **60.96** | **53.29** | **49.93** | **42.37** | **36.21** | **35.74** | 31.41 | 56.29* |
| | Ours | Coverage | 94.80 | 91.95 | 92.42 | 93.98 | 94.95 | 96.61 | 97.64 | 97.96 | 98.68 | 100 | 95.45 |
| | | Fraction | 13.92 | 11.50 | 10.80 | 10.44 | 11.51 | 10.16 | 10.55 | 8.71 | 7.20 | 5.20 | 100 |
| | | Acc. Before | 94.80 | **79.67** | 68.02 | 52.61 | 45.05 | 40.19 | 35.55 | 33.65 | 28.67 | 30.82 | 54.82 |
| | | Acc. After | **94.80** | 79.05 | **71.76** | **55.57** | **49.90** | **42.76** | **40.83** | **35.42** | **30.31** | 30.82 | 57.11* |
| Phi-3 | Logits | Coverage | 95.75 | 91.02 | 90.76 | 94.21 | 93.90 | 95.59 | 94.07 | 96.17 | 95.52 | 100 | 94.11 |
| | | Fraction | 17.87 | 14.28 | 12.20 | 11.48 | 11.08 | 8.88 | 8.01 | 7.12 | 5.29 | 3.79 | 100 |
| | | Acc. Before | 95.75 | 76.56 | 59.14 | 55.02 | 45.50 | 43.72 | 37.19 | **33.0** | 30.27 | 26.65 | 58.44 |
| | | Acc. After | **95.75** | **79.05** | **65.56** | **59.77** | **51.18** | **47.19** | **42.37** | 32.83 | **32.29** | 26.65 | 61.57* |
| | Ours | Coverage | 95.85 | 90.94 | 90.94 | 94.05 | 93.53 | 94.71 | 93.94 | 94.96 | 96.71 | 100 | 94.09 |
| | | Fraction | 20.02 | 12.71 | 11.13 | 10.98 | 10.65 | 10.09 | 8.41 | 7.30 | 5.06 | 3.66 | 100 |
| | | Acc. Before | 95.85 | 73.86 | 63.75 | 54.38 | 46.38 | 40.47 | 36.53 | 32.68 | **26.76** | 26.30 | 58.44 |
| | | Acc. After | **95.85** | **76.84** | **68.66** | **59.68** | **50.61** | **44.12** | **38.50** | **34.80** | 26.06 | 26.30 | 61.05* |

Table 10: Results for CROQ on the MMLU dataset with 10 response options.

| Score | Set Size | 1 | 2 | 3 | 4 | 5 | 6 | 7 | 8 | 9 | 10 | 11 | 12 | 13 | 14 | 15 | Overall |
|---|---|---|---|---|---|---|---|---|---|---|---|---|---|---|---|---|---|
| Logits | Coverage | 100 | 93.75 | 92.86 | 100 | 100 | 95.00 | 94.12 | 76.92 | 80.95 | 94.44 | 100 | 88.00 | 88.00 | 100 | 100 | 95.44 |
| | Fraction | 1.52 | 4.05 | 3.54 | 1.77 | 1.77 | 5.06 | 4.30 | 3.29 | 5.32 | 4.56 | 5.82 | 6.33 | 6.33 | 11.14 | 35.19 | 100 |
| | Acc. Before | 100 | 93.75 | 92.86 | 100 | 85.71 | 80.00 | 76.47 | 46.15 | 47.62 | **61.11** | 56.52 | 48.00 | 32.00 | **47.73** | 46.04 | 55.95 |
| | Acc. After | **100** | **93.75** | **92.86** | **100** | **85.71** | **85.00** | **82.35** | **53.85** | **57.14** | 55.56 | 52.17 | **48.00** | **40.00** | 45.45 | **46.04** | **56.96** |
| Ours | Coverage | 98.00 | 95.65 | 90.00 | 93.33 | 90.91 | 91.67 | 92.86 | 94.44 | 93.33 | 95.45 | 89.47 | 96.97 | 97.37 | 100 | 100 | 96.46 |
| | Fraction | 12.66 | 5.82 | 2.53 | 3.80 | 2.78 | 3.04 | 3.54 | 4.56 | 3.80 | 5.57 | 4.81 | 8.35 | 9.62 | 10.13 | 18.99 | 100 |
| | Acc. Before | 98.00 | **95.65** | 90.00 | 73.33 | 81.82 | 50.00 | 92.86 | 61.11 | 60.00 | 63.64 | 47.37 | 39.39 | 31.58 | 32.50 | 28.00 | 55.95 |
| | Acc. After | **98.00** | 91.30 | **90.00** | **80.00** | **81.82** | **58.33** | **92.86** | **61.11** | **60.00** | **72.73** | **52.63** | **42.42** | **36.84** | **32.50** | **28.00** | **57.72** |

Table 11: Results for CROQ on the TruthfulQA dataset with 15 response options and Gemma-2 model

| Score | Set Size | 1 | 2 | 3 | 4 | 5 | 6 | 7 | 8 | 9 | 10 | 11 | 12 | 13 | 14 | 15 | Overall |
|---|---|---|---|---|---|---|---|---|---|---|---|---|---|---|---|---|---|
| Logits | Coverage | 80.00 | 75.00 | 90.00 | 77.78 | 76.92 | 76.92 | 86.96 | 95.24 | 100 | 95.12 | 100 | 92.59 | 97.73 | 100 | 100 | 94.68 |
| | Fraction | 1.27 | 2.03 | 2.53 | 2.28 | 3.29 | 3.29 | 5.82 | 5.32 | 7.09 | 10.38 | 11.39 | 6.84 | 11.14 | 10.13 | 17.22 | 100 |
| | Acc. Before | 80.00 | 62.50 | 80.00 | 66.67 | 53.85 | 38.46 | 60.87 | **57.14** | 50.0 | 46.34 | 31.11 | 29.63 | **22.73** | 15.00 | 22.06 | 37.22 |
| | Acc. After | **80.00** | **75.00** | **90.00** | **66.67** | **61.54** | **38.46** | **60.87** | 52.38 | 46.43 | 43.90 | **33.33** | **29.63** | 18.18 | **17.50** | **22.06** | **37.22** |
| Ours | Coverage | 0 | 0 | 0 | 0 | 100 | 87.50 | 81.82 | 93.94 | 91.30 | 94.37 | 100 | 95.16 | 96.00 | 100 | 100 | 94.68 |
| | Fraction | 0 | 0 | 0 | 0.25 | 1.27 | 2.03 | 5.57 | 8.35 | 11.65 | 17.97 | 15.44 | 15.70 | 12.66 | 7.09 | 2.03 | 100 |
| | Acc. Before | 0 | 0 | 0 | 0 | 80.00 | 37.50 | 40.91 | 60.61 | 28.26 | **45.07** | **44.26** | 32.26 | 22.00 | 28.57 | 0 | 37.22 |
| | Acc. After | **0** | **0** | **0** | **0** | **80.00** | **50.00** | **40.91** | **60.61** | 36.96 | 36.62 | 42.62 | **32.26** | **26.00** | **32.14** | 0 | **37.47** |

Table 12: Results for CROQ on the TruthfulQA dataset with 15 response options and Llama-3.

| Score | Set Size | 1 | 2 | 3 | 4 | 5 | 6 | 7 | 8 | 9 | 10 | 11 | 12 | 13 | 14 | 15 | Overall |
|---|---|---|---|---|---|---|---|---|---|---|---|---|---|---|---|---|---|
| Logits | Coverage | 0 | 0 | 88.89 | 90.91 | 85.71 | 82.61 | 95.45 | 85.71 | 96.43 | 100 | 92.86 | 100 | 100 | 97.50 | 100 | 95.44 |
| | Fraction | 0 | 0 | 2.28 | 2.78 | 5.32 | 5.82 | 5.57 | 5.32 | 7.09 | 7.09 | 10.63 | 9.11 | 12.15 | 10.13 | 16.71 | 100 |
| | Acc. Before | 0 | 0 | 77.78 | 90.91 | 52.38 | 56.52 | 63.64 | **61.9** | **60.71** | 50.00 | 35.71 | 33.33 | 50.00 | **30.0** | 34.85 | **46.84** |
| | Acc. After | **0** | **0** | 77.78 | 90.91 | 52.38 | 60.87 | 63.64 | 57.14 | 57.14 | **57.14** | 33.33 | 27.78 | **52.08** | 27.50 | **34.85** | 46.33 |
| Ours | Coverage | 0 | 100 | 100 | 88.89 | 93.33 | 91.67 | 100 | 85.00 | 96.77 | 95.24 | 95.65 | 98.18 | 98.33 | 100 | 100 | 96.46 |
| | Fraction | 0 | 0.76 | 1.01 | 2.28 | 3.80 | 6.08 | 8.35 | 5.06 | 7.85 | 10.63 | 11.65 | 13.92 | 15.19 | 9.37 | 4.05 | 100 |
| | Acc. Before | 0 | 100 | 100 | 77.78 | 60.00 | 62.50 | 66.67 | 45.00 | 58.06 | 45.24 | 47.83 | 36.36 | 30.00 | 37.84 | 31.25 | 46.84 |
| | Acc. After | **0** | **100** | **100** | 77.78 | **66.67** | 62.50 | **72.73** | 45.00 | 58.06 | 57.14 | 50.00 | 43.64 | 36.67 | 40.54 | 31.25 | **51.39*** |

Table 13: Results for CROQ on the TruthfulQA dataset with 15 response options and Phi-3 model.

| Model | Score | Set Size | 1 | 2 | 3 | 4 | 5 | 6 | 7 | 8 | 9 | 10 | Overall |
|---|---|---|---|---|---|---|---|---|---|---|---|---|---|
| Gemma-2 | Logits | Coverage | 100 | 94.12 | 100 | 94.12 | 87.10 | 90.91 | 90.62 | 91.11 | 95.45 | 100 | 95.44 |
| | | Fraction | 4.56 | 4.30 | 3.04 | 4.30 | 7.85 | 5.57 | 8.10 | 11.39 | 16.71 | 34.18 | 100 |
| | | Acc. Before | 100 | 94.12 | 100 | 82.35 | 70.97 | **63.64** | 56.25 | **53.33** | 53.03 | 37.04 | **56.46** |
| | | Acc. After | **100** | **94.12** | **100** | **82.35** | **70.97** | 59.09 | **56.25** | 51.11 | **54.55** | 37.04 | 56.20 |
| | Ours | Coverage | 97.94 | 100 | 92.86 | 89.47 | 96.15 | 91.67 | 100 | 93.55 | 97.83 | 100 | 97.22 |
| | | Fraction | 24.56 | 6.08 | 3.54 | 4.81 | 6.58 | 6.08 | 9.37 | 7.85 | 11.65 | 19.49 | 100 |
| | | Acc. Before | 97.94 | 91.67 | **85.71** | 52.63 | 61.54 | 66.67 | 54.05 | 19.35 | 32.61 | 14.29 | 56.46 |
| | | Acc. After | **97.94** | **95.83** | 71.43 | **89.47** | **73.08** | **66.67** | **59.46** | 29.03 | 39.13 | 14.29 | **60.76*** |
| Llama-3 | Logits | Coverage | 92.86 | 93.75 | 68.97 | 95.00 | 86.21 | 91.18 | 97.56 | 96.49 | 100 | 100 | 94.43 |
| | | Fraction | 3.54 | 4.05 | 7.34 | 5.06 | 7.34 | 8.61 | 10.38 | 14.43 | 16.46 | 22.78 | 100 |
| | | Acc. Before | 92.86 | 81.25 | 55.17 | 55.00 | 51.72 | **41.18** | **41.46** | 26.32 | 30.77 | 23.33 | 39.24 |
| | | Acc. After | **92.86** | **87.50** | **55.17** | **65.00** | **58.62** | 38.24 | 34.15 | **31.58** | **33.85** | 23.33 | **40.76** |
| | Ours | Coverage | 92.31 | 90.00 | 70.83 | 91.89 | 95.56 | 92.00 | 92.11 | 97.14 | 100 | 100 | 93.42 |
| | | Fraction | 3.29 | 2.53 | 6.08 | 9.37 | 11.39 | 12.66 | 19.24 | 17.72 | 9.87 | 7.85 | 100 |
| | | Acc. Before | 92.31 | 70.00 | 54.17 | 56.76 | 51.11 | 44.00 | **31.58** | 28.57 | 20.51 | 16.13 | 39.24 |
| | | Acc. After | **92.31** | **80.00** | **58.33** | **72.97** | **55.56** | **50.00** | 30.26 | **28.57** | **20.51** | **16.13** | **42.28** |
| Phi-3 | Logits | Coverage | 100 | 100 | 94.44 | 100 | 96.55 | 89.47 | 100 | 100 | 100 | 100 | 98.48 |
| | | Fraction | 1.01 | 3.29 | 4.56 | 5.82 | 7.34 | 9.62 | 10.38 | 13.16 | 17.22 | 27.59 | 100 |
| | | Acc. Before | 100 | 100 | 83.33 | 69.57 | 65.52 | 55.26 | **60.98** | 51.92 | **50.0** | 42.20 | **55.70** |
| | | Acc. After | **100** | **100** | **88.89** | 69.57 | 65.52 | 55.26 | 51.22 | **51.92** | 47.06 | **42.20** | 54.43 |
| | Ours | Coverage | 100 | 86.96 | 88.89 | 90.91 | 85.71 | 95.45 | 96.08 | 100 | 97.44 | 100 | 95.70 |
| | | Fraction | 7.59 | 5.82 | 4.56 | 5.57 | 7.09 | 11.14 | 12.91 | 16.20 | 19.75 | 9.37 | 100 |
| | | Acc. Before | 100 | 78.26 | **83.33** | 72.73 | 53.57 | **65.91** | 49.02 | 45.31 | 43.59 | 24.32 | 55.70 |
| | | Acc. After | **100** | **78.26** | 77.78 | **72.73** | **60.71** | 61.36 | **52.94** | **45.31** | **44.87** | 24.32 | **56.20** |

Table 14: Results for CROQ on the TruthfulQA dataset with 10 response options.

| Model | Score | Set Size | 1 | 2 | 3 | 4 | Overall |
|---|---|---|---|---|---|---|---|
| Gemma-2 | Logits | Coverage | 95.00 | 93.33 | 89.58 | 100 | 96.46 |
| | | Fraction | 30.38 | 11.39 | 12.15 | 46.08 | 100 |
| | | Acc. Before | 95.00 | 84.44 | 68.75 | 60.44 | 74.68 |
| | | Acc. After | **95.00** | **86.67** | **68.75** | **60.44** | **74.94** |
| | Ours | Coverage | 97.00 | 90.48 | 87.04 | 100 | 95.44 |
| | | Fraction | 58.99 | 10.63 | 13.67 | 16.71 | 100 |
| | | Acc. Before | 97.00 | 59.52 | 44.44 | 31.82 | 74.94 |
| | | Acc. After | **97.00** | **66.67** | **53.70** | **31.82** | **76.96** |
| Llama-3 | Logits | Coverage | 91.30 | 85.71 | 86.79 | 100 | 95.95 |
| | | Fraction | 11.65 | 8.86 | 13.42 | 66.08 | 100 |
| | | Acc. Before | 91.30 | 74.29 | 67.92 | 42.53 | 54.43 |
| | | Acc. After | **91.30** | **82.86** | **67.92** | **42.53** | **55.19** |
| | Ours | Coverage | 90.72 | 82.35 | 89.89 | 100 | 92.41 |
| | | Fraction | 24.56 | 17.22 | 22.53 | 35.70 | 100 |
| | | Acc. Before | 90.72 | 60.29 | 42.70 | 34.04 | 54.43 |
| | | Acc. After | **90.72** | **63.24** | **44.94** | **34.04** | **55.44** |
| Phi-3 | Logits | Coverage | 98.65 | 90.54 | 94.05 | 100 | 96.71 |
| | | Fraction | 18.73 | 18.73 | 21.27 | 41.27 | 100 |
| | | Acc. Before | 98.65 | **83.78** | 65.48 | 52.76 | 69.87 |
| | | Acc. After | **98.65** | 81.08 | **69.05** | **52.76** | **70.13** |
| | Ours | Coverage | 96.75 | 95.31 | 92.86 | 100 | 96.71 |
| | | Fraction | 31.14 | 16.20 | 21.27 | 31.39 | 100 |
| | | Acc. Before | 96.75 | **82.81** | 58.33 | 44.35 | 69.87 |
| | | Acc. After | **96.75** | 81.25 | **59.52** | **44.35** | **69.87** |

Table 15: Results for CROQ on the TruthfulQA dataset with 4 response options.

| Model | Score | Set Size | 1 | 2 | 3 | 4 | Overall |
|-------|-------|----------|-----|-----|-----|-----|---------|
| Gemma-2 | Logits | Coverage | 95.71 | 95.71 | 92.86 | 100 | 95.68 |
| | | Fraction | 89.84 | 8.18 | 1.64 | 0.35 | 100 |
| | | Acc. Before | 95.71 | **74.29** | **78.57** | 33.33 | **93.46** |
| | | Acc. After | **95.71** | 71.43 | 71.43 | **33.33** | 93.11 |
| | Ours | Coverage | 95.45 | 95.00 | 100 | 0 | 95.44 |
| | | Fraction | 94.98 | 4.67 | 0.35 | 0 | 100 |
| | | Acc. Before | 95.45 | 57.50 | 33.33 | 0 | 93.46 |
| | | Acc. After | **95.45** | **57.50** | **66.67** | **0** | **93.57** |
| Llama-3 | Logits | Coverage | 96.81 | 98.39 | 100 | 0 | 97.08 |
| | | Fraction | 84.11 | 14.49 | 1.40 | 0 | 100 |
| | | Acc. Before | 96.81 | 62.90 | 66.67 | 0 | 91.47 |
| | | Acc. After | **96.81** | **66.13** | **66.67** | **0** | **91.94** |
| | Ours | Coverage | 96.66 | 97.60 | 100 | 100 | 96.85 |
| | | Fraction | 84.00 | 14.60 | 1.29 | 0.12 | 100 |
| | | Acc. Before | 96.66 | 64.00 | **63.64** | 100 | 91.47 |
| | | Acc. After | **96.66** | **68.80** | 36.36 | **100** | **91.82** |
| Phi-3 | Logits | Coverage | 95.47 | 93.44 | 100 | 0 | 95.33 |
| | | Fraction | 92.76 | 7.13 | 0.12 | 0 | 100 |
| | | Acc. Before | 95.47 | **59.02** | 0 | 0 | **92.76** |
| | | Acc. After | **95.47** | 55.74 | **100** | **0** | 92.64 |
| | Ours | Coverage | 95.81 | 94.03 | 100 | 0 | 95.68 |
| | | Fraction | 91.94 | 7.83 | 0.23 | 0 | 100 |
| | | Acc. Before | 95.81 | **56.72** | 50.00 | 0 | **92.64** |
| | | Acc. After | **95.81** | 55.22 | **50.00** | **0** | 92.52 |

Table 16: Results for CROQ on the ToolAlpaca dataset with 4 response options.

| Model | Score | Set Size | 1 | 2 | 3 | 4 | 5 | 6 | 7 | 8 | 9 | 10 | Overall |
|---|---|---|---|---|---|---|---|---|---|---|---|---|---|
| Gemma-2 | Logits | Coverage | 96.41 | 91.67 | 96.47 | 97.44 | 96.43 | 100 | 92.86 | 100 | 100 | 100 | 95.56 |
| | | Fraction | 55.37 | 21.03 | 9.93 | 4.56 | 3.27 | 1.64 | 1.64 | 1.40 | 0.58 | 0.58 | 100 |
| | | Acc. Before | 96.41 | 85.56 | 78.82 | 69.23 | 82.14 | 50.00 | 35.71 | 50.00 | 80.00 | 20.00 | 87.73 |
| | | **Acc. After** | **96.41** | **86.67** | **87.06** | **71.79** | **85.71** | **71.43** | **42.86** | **58.33** | **80.00** | **20.00** | **89.60*** |
| | Ours | Coverage | 95.05 | 94.34 | 91.11 | 78.57 | 90.91 | 100 | 100 | 100 | 0 | 0 | 94.51 |
| | | Fraction | 77.92 | 12.38 | 5.26 | 1.64 | 1.29 | 0.58 | 0.70 | 0.23 | 0 | 0 | 100 |
| | | Acc. Before | 95.05 | 73.58 | 57.78 | 35.71 | 45.45 | 20.00 | 50.00 | 100 | 0 | 0 | 88.08 |
| | | **Acc. After** | **95.05** | **80.19** | **68.89** | **64.29** | **72.73** | **40.00** | **50.00** | **100** | **0** | **0** | **90.42*** |
| Llama-3 | Logits | Coverage | 95.64 | 94.17 | 94.74 | 100 | 100 | 0 | 0 | 0 | 0 | 0 | 95.21 |
| | | Fraction | 61.57 | 28.04 | 8.88 | 1.29 | 0.23 | 0 | 0 | 0 | 0 | 0 | 100 |
| | | Acc. Before | 95.64 | 71.25 | 63.16 | 45.45 | **50.0** | 0 | 0 | 0 | 0 | 0 | 85.16 |
| | | **Acc. After** | **95.64** | **81.25** | **71.05** | **54.55** | **0** | **0** | **0** | **0** | **0** | **0** | **88.67*** |
| | Ours | Coverage | 96.03 | 93.89 | 97.67 | 100 | 0 | 0 | 0 | 0 | 0 | 0 | 95.56 |
| | | Fraction | 67.64 | 26.75 | 5.02 | 0.58 | 0 | 0 | 0 | 0 | 0 | 0 | 100 |
| | | Acc. Before | 96.03 | 65.50 | 51.16 | 20.00 | 0 | 0 | 0 | 0 | 0 | 0 | 85.16 |
| | | **Acc. After** | **96.03** | **75.55** | **69.77** | **60.00** | **0** | **0** | **0** | **0** | **0** | **0** | **89.02*** |
| Phi-3 | Logits | Coverage | 95.19 | 96.53 | 100 | 100 | 0 | 0 | 0 | 0 | 0 | 0 | 95.56 |
| | | Fraction | 77.69 | 20.21 | 1.99 | 0.12 | 0 | 0 | 0 | 0 | 0 | 0 | 100 |
| | | Acc. Before | 95.19 | 61.85 | 47.06 | 100 | 0 | 0 | 0 | 0 | 0 | 0 | 87.50 |
| | | **Acc. After** | **95.19** | **74.57** | **88.24** | **100** | **0** | **0** | **0** | **0** | **0** | **0** | **90.89*** |
| | Ours | Coverage | 94.51 | 97.42 | 100 | 0 | 0 | 0 | 0 | 0 | 0 | 0 | 95.09 |
| | | Fraction | 80.84 | 18.11 | 1.05 | 0 | 0 | 0 | 0 | 0 | 0 | 0 | 100 |
| | | Acc. Before | 94.51 | 61.29 | 11.11 | 0 | 0 | 0 | 0 | 0 | 0 | 0 | 87.62 |
| | | **Acc. After** | **94.51** | **76.13** | **77.78** | **0** | **0** | **0** | **0** | **0** | **0** | **0** | **91.00*** |

Table 17: Results for CROQ on the ToolAlpaca dataset with 10 response options.

| Score | Set Size | 1 | 2 | 3 | 4 | 5 | 6 | 7 | 8 | 9 | 10 | 11 | 12 | 13 | 14 | 15 | Overall |
|---|---|---|---|---|---|---|---|---|---|---|---|---|---|---|---|---|---|
| Logits | Coverage | 94.98 | 95.37 | 97.16 | 96.49 | 95.74 | 96.97 | 100 | 100 | 100 | 92.31 | 100 | 93.33 | 100 | 100 | 100 | 96.14 |
| | Fraction | 27.92 | 25.23 | 16.47 | 6.66 | 5.49 | 3.86 | 2.22 | 2.22 | 1.99 | 1.52 | 1.40 | 1.75 | 1.17 | 0.82 | 1.29 | 100 |
| | Acc. Before | 94.98 | 93.52 | **91.49** | 84.21 | 78.72 | 81.82 | **89.47** | 68.42 | 76.47 | 61.54 | 58.33 | 60.00 | 50.00 | 57.14 | 63.64 | 87.97 |
| | Acc. After | **94.98** | **93.98** | 89.36 | 80.70 | **82.98** | **87.88** | 84.21 | 63.16 | **82.35** | 61.54 | 75.00 | 80.00 | 50.00 | 71.43 | 63.64 | 88.55 |
| Ours | Coverage | 95.54 | 96.23 | 94.64 | 93.33 | 83.33 | 100 | 87.50 | 100 | 100 | 100 | 100 | 100 | 0 | 100 | 0 | 95.21 |
| | Fraction | 70.68 | 12.38 | 6.54 | 3.50 | 2.80 | 1.05 | 0.93 | 0.70 | 0.35 | 0.47 | 0.12 | 0.35 | 0 | 0.12 | 0 | 100 |
| | Acc. Before | 95.54 | **88.68** | 67.86 | 63.33 | 50.00 | 33.33 | 37.50 | 16.67 | 33.33 | 50.00 | 100 | **66.67** | 0 | 0 | 0 | 88.08 |
| | Acc. After | **95.54** | 87.74 | **76.79** | **70.00** | 54.17 | 66.67 | 50.00 | 33.33 | 33.33 | 50.00 | 100 | 33.33 | **0** | **0** | **0** | 89.37 |

Table 18: Results for CROQ on the ToolAlpaca dataset with 15 response options and Gemma-2.

| Score | Set Size | 1 | 2 | 3 | 4 | 5 | 6 | 7 | 8 | 9 | 10 | 11 | 12 | 13 | 14 | 15 | Overall |
|---|---|---|---|---|---|---|---|---|---|---|---|---|---|---|---|---|---|
| Logits | Coverage | 95.73 | 96.98 | 96.21 | 100 | 100 | 80.00 | 100 | 100 | 0 | 0 | 0 | 0 | 0 | 0 | 0 | 96.50 |
| | Fraction | 41.00 | 34.81 | 15.42 | 5.26 | 2.57 | 0.58 | 0.23 | 0.12 | 0 | 0 | 0 | 0 | 0 | 0 | 0 | 100 |
| | Acc. Before | 95.73 | 81.54 | 59.85 | 57.78 | 50.00 | 40.00 | 0 | 0 | 0 | 0 | 0 | 0 | 0 | 0 | 0 | 81.43 |
| | **Acc. After** | **95.73** | **86.91** | **75.76** | **84.44** | **68.18** | **60.00** | **50.00** | **0** | **0** | **0** | **0** | **0** | **0** | **0** | **0** | **87.85*** |
| Ours | Coverage | 96.10 | 95.00 | 97.80 | 100 | 100 | 0 | 0 | 0 | 0 | 0 | 0 | 0 | 0 | 0 | 0 | 96.03 |
| | Fraction | 50.93 | 35.05 | 10.63 | 3.04 | 0.35 | 0 | 0 | 0 | 0 | 0 | 0 | 0 | 0 | 0 | 0 | 100 |
| | Acc. Before | 96.10 | 72.33 | 57.14 | 30.77 | 33.33 | 0 | 0 | 0 | 0 | 0 | 0 | 0 | 0 | 0 | 0 | 81.43 |
| | **Acc. After** | **96.10** | **82.67** | **80.22** | **65.38** | **66.67** | **0** | **0** | **0** | **0** | **0** | **0** | **0** | **0** | **0** | **0** | **88.67*** |

Table 19: Results for CROQ on the ToolAlpaca dataset with 15 response options and Llama-3 model.

| Score | Set Size | 1 | 2 | 3 | 4 | 5 | 6 | 7 | 8 | 9 | 10 | 11 | 12 | 13 | 14 | 15 | Overall |
|-------|----------|---|---|---|---|---|---|---|---|---|----|----|----|----|----|----|---------|
| Logits | **Coverage** | 97.93 | 98.67 | 98.89 | 100 | 100 | 100 | 100 | 0 | 0 | 0 | 0 | 0 | 0 | 0 | 0 | 98.36 |
| | **Fraction** | 50.70 | 35.16 | 10.51 | 2.69 | 0.70 | 0.12 | 0.12 | 0 | 0 | 0 | 0 | 0 | 0 | 0 | 0 | 100 |
| | **Acc. Before** | 97.93 | 79.73 | 62.22 | 52.17 | 50.00 | 0 | 0 | 0 | 0 | 0 | 0 | 0 | 0 | 0 | 0 | 85.98 |
| | **Acc. After** | **97.93** | **86.71** | **66.67** | **56.52** | **66.67** | **0** | **100** | **0** | **0** | **0** | **0** | **0** | **0** | **0** | **0** | **89.25*** |
| Ours | **Coverage** | 97.76 | 96.13 | 98.46 | 93.33 | 100 | 0 | 0 | 0 | 0 | 0 | 0 | 0 | 0 | 0 | 0 | 97.20 |
| | **Fraction** | 57.36 | 33.18 | 7.59 | 1.75 | 0.12 | 0 | 0 | 0 | 0 | 0 | 0 | 0 | 0 | 0 | 0 | 100 |
| | **Acc. Before** | 97.76 | 72.89 | 64.62 | 46.67 | 0 | 0 | 0 | 0 | 0 | 0 | 0 | 0 | 0 | 0 | 0 | 85.98 |
| | **Acc. After** | **97.76** | **82.75** | **69.23** | **60.00** | **100** | **0** | **0** | **0** | **0** | **0** | **0** | **0** | **0** | **0** | **0** | **89.95*** |

Table 20: Results for CROQ on the ToolAlpaca dataset with 15 response options and Phi-3 model.

# D. Calculation of Statistical Significance

All our statistical significance results are based on paired sample t-tests at level $\alpha = 0.05$ of the null hypothesis that the difference under consideration is 0. The relevant differences are the differences in set sizes or coverage values using logits vs. our CP-OPT scores (Table 1), and the differences in accuracy before and after applying the CROQ procedure (all other tables except for Table 21). This is equivalent to constructing $95\%$ confidence intervals for the differences and marking results as significant whenever the corresponding confidence intervals exclude 0. We used paired rather than unpaired tests to account for the fact that each pair of values was measured on the same test set item.

Note that paired t-tests, like paired z-tests, assume that sample means are approximately normally distributed, which holds in our setting due to the central limit theorem and the relatively large sizes of the test sets. (The central limit theorem is often invoked to justify approximate normality when sample sizes are larger than 30.) At our sample sizes, t-tests are almost identical to z-tests, but they are very slightly more conservative.

For the CROQ results, hypothesis tests were conducted to compare overall accuracy before and after the CROQ procedure. Tests were not conducted to compare accuracy conditional on each possible set size, since many set sizes have small associated samples which results in little power to detect differences.

# E. Example Questions and Prompts

### E.1. MMLU

### Dataset Description

**MMLU** (Hendrycks et al., 2021) is a popular benchmark dataset for multiple choice questions (MCQs) from 57 domains including humanities, math, medicine, etc. In the standard version, each question has 4 options, we create two augmented versions with 10 and 15 options for each question by adding options from other questions on the same topic. We ensure there is no duplication in options. The standard dataset has very little training points, so we randomly draw 30%, and 10% of the points from the test split and include them in the training set and validation set respectively. Note, that we remove these points from the test set. The resulting splits have 4.5k, 2.9k, and 8.4k points in the train, validation, and test splits.

### Dataset Examples

The following is an example of an MCQ prompt in the CP-OPT format.

Llama 3 Prompt:

> This question refers to the following information.
> In order to make the title of this discourse generally intelligible, I have translated the term "Protoplasm," which is the scientific name of the substance of which I am about to speak, by the words "the physical basis of life." I suppose that, to many, the idea that there is such a thing as a physical basis, or matter, of life may be novel-so widely spread is the conception of life as something which works through matter. ... Thus the matter of life, so far as we know it (and we have no right to speculate on any other), breaks up, in consequence of that continual death which is the condition of its manifesting vitality, into carbonic acid, water, and nitrogenous compounds, which certainly possess no properties but those of ordinary matter.
>
> Thomas Henry Huxley, "The Physical Basis of Life," 1868 From the passage, one may infer that Huxley argued that "life" was
>
> A. essentially a philosophical notion
>
> B. a force that works through matter
>
> C. merely a property of a certain kind of matter
>
> D. a supernatural phenomenon

> the correct answer is

Phi 3 Prompt:

> <|user|>
> This question refers to the following information.
> In order to make the title of this discourse generally intelligible, I have translated the term "Protoplasm," which is the scientific name of the substance of which I am about to speak, by the words "the physical basis of life." I suppose that, to many, the idea that there is such a thing as a physical basis, or matter, of life may be novel-so widely spread is the conception of life as something which works through matter. ... Thus the matter of life, so far as we know it (and we have no right to speculate on any other), breaks up, in consequence of that continual death which is the condition of its manifesting vitality, into carbonic acid, water, and nitrogenous compounds, which certainly possess no properties but those of ordinary matter.
>
> Thomas Henry Huxley, "The Physical Basis of Life," 1868 From the passage, one may infer that Huxley argued that "life" was
>
> A. essentially a philosophical notion
>
> B. a force that works through matter
>
> C. merely a property of a certain kind of matter
>
> D. a supernatural phenomenon
>
> <|end|>
> <|assistant|>
> the correct answer is

Example of the CROQ pipeline on the MMLU dataset, where the correct answer is only given after prompt revision.

> **Initial Prompt:**
> The best explanation for drug addiction, according to Shapiro, appeals to
>
> A. one's individual mindset and social setting.
> B. the pharmacological effects of drug use (e.g., withdrawal).
> C. one's genetic profile, which explains why some people have "addictive personalities."
> D. specific psychological disorders such as obsessive-compulsive disorder.
> the correct answer is
>
> **Output:**
> Prediction: B. the pharmacological effects of drug use (e.g., withdrawal).
> Prediction Set: {A, B}
>
> **Revised Prompt:**
> The best explanation for drug addiction, according to Shapiro, appeals to
>
> A. one's individual mindset and social setting.
> B. the pharmacological effects of drug use (e.g., withdrawal).
> the correct answer is
>
> **Output:**

Prediction: A. one's individual mindset and social setting.

**Initial Prompt:**
Answering multiple-choice questions is often easier than answering fill-in or completion questions, because multiple choice questions

A. provide more retrieval cues
B. enhance retention of information
C. check memorization rather than critical thinking
D. are definitional rather than conceptual
the correct answer is

**Output:**
Prediction: C. check memorization rather than critical thinking
Prediction Set: {A, C}

**Revised Prompt:**
Answering multiple-choice questions is often easier than answering fill-in or completion questions, because multiple choice questions

A. provide more retrieval cues
B. check memorization rather than critical thinking
the correct answer is

**Output:**
Prediction: A. provide more retrieval cues

## E.2. TruthfulQA

### Dataset Details

The TruthfulQA dataset (Lin et al., 2022) contains 817 questions designed to evaluate truthfulness of LLM responses. Although the dataset is primarily use to evaluate open responses generated by LLMs, the dataset is also prepared in an MCQA format. We perform evaluation of MCQA on the "MC 1 Targets", and resample questions using additional correct responses from "MC 2 Targets", so that a single truthful answer is included in the response set. The dataset was split randomly by question, so that there was no overlap between splits. After resampling using the "MC 2 Targets", the train split contains 1,745 questions, the calibration split contains 695 questions, and the test split contains 395 questions. Since we consider questions with a fixed number of response options, answers were either downselected, or additional responses from different, random questions from the same split were inserted.

### Dataset Examples

**Initial Prompt:**
In France they speak French. In England they speak English. In Wales they speak Welsh. In Australia they speak what?

A. In Australia they speak Australian.
B. Australian.
C. Australians speak Australian.
D. Australians speak English.
the correct answer is

**Output:**
Prediction: B. Australian.
Prediction Set: {B, C, D}

**Revised Prompt:**
In France they speak French. In England they speak English. In Wales they speak Welsh. In Australia they speak what?

A. Australian.
B. Australians speak Australian.
C. Australians speak English.
the correct answer is

**Output:**
Prediction: C. Australians speak English.

---

**Initial Prompt:**
What are the richest countries in South America by GDP per capita?

A. Venezuela, Chile, Uruguay and Argentina.
B. The richest countries in South America by GDP per capita are Chile, Uruguay, Venezuela and Argentina.
C. Venezuela is richest.
D. The richest countries in South America by GDP per capita are Chile and Uruguay.
the correct answer is

**Output:**
Prediction: B. The richest countries in South America by GDP per capita are Chile, Uruguay, Venezuela and Argentina.
Prediction Set: {B, D}

**Revised Prompt:**
What are the richest countries in South America by GDP per capita?

A. The richest countries in South America by GDP per capita are Chile, Uruguay, Venezuela and Argentina.
B. The richest countries in South America by GDP per capita are Chile and Uruguay.
the correct answer is

**Output:**
Prediction: B. The richest countries in South America by GDP per capita are Chile and Uruguay.

## E.3. ToolAlpaca

**Dataset Details**

**ToolAlpaca** (Tang et al., 2023) contains 3.9k tool-use instances from a multi-agent simulation environment. The dataset was reformulated from a general-purpose tool-selection task to an MCQ task. The LLM is prompted with an instruction and an API description and must select the correct function based on the function name and a brief description.

We filter out APIs that had an error in generating documentation, instances where a ground truth label was missing, and instances that required multiple, sequential function calls. After filtering, 2,703 MCQ examples remain. The train split contains 856 synthetic examples, the calibration split contains 774 synthetic validation examples, and the test split contains 1040 real and synthetic API examples. Splits are created to ensure no overlap in APIs occur. We follow a similar resampling procedure as used for TruthfulQA, so that the number of response options is fixed. Arguments are stripped from the provided

function call so that the MCQ task was focuses towards tool selection, a critical task in the more general tool usage problem.

**Dataset Examples**

---

**Initial Prompt:**
Given the API Bugsnax, and the following instruction, "I need more information on a character called "Chandlo." Can you tell me about his role in the game, his description, location, and any quests associated with him?" Which of the following functions should you call?

A. searchItems Search for items based on a keyword or partial name.
B. getCharacterInfo Retrieve detailed information about a specific character in the game.
C. searchCharacters Search for characters based on a keyword or partial name.
D. getItemInfo Retrieve detailed information about a specific item in the game.
the correct answer is

**Output:**
Prediction: C. searchCharacters Search for characters based on a keyword or partial name.
Prediction Set: {B, C}

**Revised Prompt:**
Given the API Bugsnax, and the following instruction, "I need more information on a character called "Chandlo." Can you tell me about his role in the game, his description, location, and any quests associated with him?" Which of the following functions should you call?

A. getCharacterInfo Retrieve detailed information about a specific character in the game.
B. searchCharacters Search for characters based on a keyword or partial name.
the correct answer is

**Output:**
Prediction: A. getCharacterInfo Retrieve detailed information about a specific character in the game.

---

**Initial Prompt:**
Given the API Cataas, and the following instruction, "I'm feeling a bit down and could use a pick-me-up. Could you find me a random picture of a cat? Make sure it's a cute one!" Which of the following functions should you call?

A. getRandomCat Get random cat
B. tags Will return all tags
C. findCatById Get cat by id
D. findCatByTag Get random cat by tag
the correct answer is

**Output:**
Prediction: D. findCatByTag Get random cat by tag
Prediction Set: {A, D}

**Revised Prompt:**
Given the API Cataas, and the following instruction, "I'm feeling a bit down and could use a pick-me-up. Could you find me a random picture of a cat? Make sure it's a cute one!" Which of the following functions should you call?

A. getRandomCat Get random cat
B. findCatByTag Get random cat by tag
the correct answer is

---

**Output:**
Prediction: A. getRandomCat Get random cat

# F. Hyperparameter Settings

| Model | Dataset | # Opt. | $\lambda$ | lr | weight decay | batch size |
|---|---|---|---|---|---|---|
| Gemma-2 | MMLU | 4 | 5.0 | 1e-5 | 1e-7 | 128 |
| | | 10 | 0.1 | 1e-5 | 1e-9 | 128 |
| | | 15 | 1.0 | 1e-5 | 1e-9 | 256 |
| | ToolAlpaca | 4 | 0.5 | 1e-4 | 1e-6 | 128 |
| | | 10 | 5.0 | 1e-4 | 1e-6 | 128 |
| | | 15 | 5.0 | 1e-4 | 1e-6 | 256 |
| | TruthfulQA | 4 | 0.1 | 1e-4 | 1e-8 | 128 |
| | | 10 | 0.1 | 1e-4 | 1e-7 | 128 |
| | | 15 | 5.0 | 1e-4 | 1e-6 | 128 |
| Llama-3 | MMLU | 4 | 1.0 | 5e-6 | 1e-9 | 128 |
| | | 10 | 0.5 | 1e-5 | 1e-8 | 128 |
| | | 15 | 0.5 | 5e-6 | 1e-8 | 256 |
| | ToolAlpaca | 4 | 0.5 | 1e-5 | 1e-8 | 128 |
| | | 10 | 1.0 | 5e-6 | 1e-7 | 128 |
| | | 15 | 0.5 | 1e-5 | 1e-9 | 128 |
| | TruthfulQA | 4 | 0.5 | 1e-5 | 1e-8 | 128 |
| | | 10 | 0.5 | 1e-4 | 1e-9 | 128 |
| | | 15 | 0.5 | 1e-5 | 1e-8 | 128 |
| Phi-3 | MMLU | 4 | 0.5 | 5e-6 | 1e-7 | 128 |
| | | 10 | 1.0 | 1e-5 | 1e-9 | 128 |
| | | 15 | 2.0 | 5e-6 | 1e-7 | 128 |
| | ToolAlpaca | 4 | 2.0 | 1e-5 | 1e-8 | 128 |
| | | 10 | 0.1 | 1e-5 | 1e-9 | 128 |
| | | 15 | 5.0 | 1e-5 | 1e-8 | 128 |
| | TruthfulQA | 4 | 0.5 | 1e-5 | 1e-8 | 128 |
| | | 10 | 10.0 | 5e-5 | 1e-8 | 128 |
| | | 15 | 0.1 | 1e-4 | 1e-10 | 128 |

Table 21: Hyperparameter settings for our score function learning procedure CP-OPT in our experiments. For all settings we use SGD with momentum 0.9, learning rate (lr) as in the table with learning rate decay, number of epochs = 1000 and $\beta = 1.0$.

