# OpenReview forum: "Prune 'n Predict: Optimizing LLM Decision-making with Conformal Prediction"
_ICML.cc/2025/Conference — ICML 2025 poster_

### Official Review · Reviewer_97WN · 2025-03-10

**Overall Recommendation:** 5

**Summary:**

The paper proposes conformal revision of questions (CROQ), which revises multiple-choice questions (MCQs) by narrowing down the prediction set. Additionally, the paper provides a corresponding routine for learning a CP score that aims to minimize set size under the coverage constraint. The experiments of the paper cover MMLU, ToolAlpaca, and TruthfulQA, using multiple LLM models. The results show CROQ improves accuracy of test-time inference.

**Claims And Evidence:**

* Claims are clearly stated in experiments section and are supported empirically.
* Additionally, the basic premise that pruning response options leads to an improvement in accuracy is nicely shown in Figure 1.

**Essential References Not Discussed:**

* I think this has been covered as far as I can tell.

**Experimental Designs Or Analyses:**

* No issue here.

**Methods And Evaluation Criteria:**

* No issue here.

**Other Comments Or Suggestions:**

* See below.

**Other Strengths And Weaknesses:**

### Strengths
* The paper is very well-structured. It was made clear early in the paper what the idea of the work was and how it was going to be achieved. The description of the relevant topics such as CP was nicely explained at a level suitable for the paper. Sometimes CP can be poorly explained and overcomplicated.
* The idea to prune based on the prediction sets is a strength.
* Experimental design, including clear hypotheses and statistical significant tests are a strength.
### Minor Weakness
* The experimental results seem to point to a side effect, (seen in Figures 4 and 5), where when there are fewer choices to select from at the start, it seems like the utility of CROQ might result in worse accuracy. This can be seen from the MMLU-4 and ToolAlpaca-4 data sets.

**Questions For Authors:**

* Was there any particular reason that $g(\cdot)$ used $tanh$ nonlinearities?
* Is the effectiveness of CP-OPT more dependent on whether the distribution of the training set matches the testing set compared to using the logits? Are there any results in the paper that might show this behaviour?

**Relation To Broader Scientific Literature:**

* The main contribution of the paper to the literature appears to be the idea of pruning answers according to the prediction sets. The secondary contribution is in the learning of a function for the conformity score. I am not aware of previous work that has used CP for pruning.

**Theoretical Claims:**

* No issue here.

---

> ### Author Rebuttal · Authors · 2025-04-01
>
> We are delighted with the positive feedback on our paper. We appreciate the recognition of strengths on *the ideas, presentation and soundness of empirical analysis*. Our response to the queries is as follows.
>
> **Why tanh for $g$ ?** While the choice of the function class $\mathcal{G}$ is up to the user, in general it makes sense to use a flexible non-linear function class. A multi-layer neural network with any activation function could be a good fit here. We chose tanh as its range is (-1, 1) and is known to have nice properties such as resulting in efficient training [1].
>
> [1] https://cseweb.ucsd.edu/classes/wi08/cse253/Handouts/lecun-98b.pdf
>
> **On (mis)match between distribution of training and test set.**  In this work we assume that the training, calibration and test data are i.i.d. (independent and identically distributed). Logit scores use calibration data to estimate the threshold for conformal prediction, which is used on the test data to create prediction sets. Thus, we evaluate both scores under the same conditions on the data.
>
> If we anticipate distribution shift in the test set, we could modify the CP-OPT objective to use distributionally robust optimization (DRO) techniques, so that it is robust to distribution shifts. This is beyond the scope of our work and could be interesting to explore in future.
>
> We hope our response resolves the queries. We are happy to answer any further questions you may have.

---

### Official Review · Reviewer_uJFE · 2025-03-14

**Overall Recommendation:** 2

**Summary:**

This paper proposes a method to enhance large language model (LLM) decision-making for multiple-choice questions (MCQs) and tool selection tasks using conformal prediction (CP). The authors introduce "Conformal Revision of Questions" (CROQ), which uses CP to identify and eliminate unlikely answer choices before re-prompting the LLM with the reduced set of options. They demonstrate that LLMs perform better when presented with fewer choices. Additionally, they propose CP-OPT, a score optimization framework that learns custom scoring functions to minimize prediction set sizes while maintaining statistical coverage guarantees.

**Claims And Evidence:**

The fundamental claim that reducing answer choices improves LLM accuracy is well-supported in Figure 1. This is exactly the case when we want to optimize tool use.
However, the claim that CP-OPT produces smaller prediction sets than logit scores and that CP-OPT scores outperform logit scores when used with CROQ are not well-supported by experiments. The improvement over logic scores is often quite limited.

**Essential References Not Discussed:**

The authors only talks about conformal prediction for LLMs in related works and lack important literatures of UQ and confidence callibration.

**Experimental Designs Or Analyses:**

1. For MMLU, the authors created versions with 10 and 15 options by adding options from other questions on the same topic. This artificial augmentation may not reflect natural MCQ distributions and could introduce biases.
2. The method should also compare with different score functions like UQ methods.

**Methods And Evaluation Criteria:**

In general, the method makes sense for MCQ settings.

There are several issues:
1. The baseline comparison is limited to logit scores from LLMs. Comparison with other uncertainty quantification methods for LLMs is needed.
2. The selection of miscoverage rate α is arbitrary (set at 0.05 for most experiments). It could be better to provide clear guidance on selecting α.

**Other Comments Or Suggestions:**

1. runtime and computational resource requirements should be analyzed
2. typos. e.g. "strengthens" in section 6

**Other Strengths And Weaknesses:**

Strength:
The idea of using conformal prediction to revise MCQs is clear to me.
The approach requires no fine-tuning, making it broadly applicable.

Weakness:
1. The improvement is often limited. And the author should compare with other prompting methods like CoT, self-refine, etc.
2. This paper focuses solely on MCQ and tool selection tasks. In practical applications, open-ended QA tasks are much more common and often more valuable than multiple-choice formats. The CROQ method, which relies on pruning answer choices through conformal prediction, is fundamentally designed for settings with discrete, pre-defined answer options. This approach cannot be directly applied to open-ended QA tasks where the space of possible answers is effectively infinite.

**Questions For Authors:**

1. How does the computational cost of CP-OPT compare to alternatives like self-consistency methods?
2. The improvements from CP-OPT over logits seem modest in many settings. Can you provide deeper insights into when and why CP-OPT significantly outperforms logits?
3. How does CROQ perform when combined with other LLM performance enhancement techniques like chain-of-thought, CoT SC, few-shot?

**Relation To Broader Scientific Literature:**

The paper adequately situates itself within the conformal prediction and LLM uncertainty quantification literature. It builds on prior work applying conformal prediction to LLMs and extends this to downstream MCQ task improvement.

**Theoretical Claims:**

There's no theoretical analysis of the convergence properties of the optimization procedure or guarantees that the learned score functions will approach the optimal ones.
While the authors correctly cite the standard coverage guarantee for split conformal prediction in Proposition 2.1, they don't provide theoretical analysis of how their specific implementation might affect this guarantee in this setting or in practice.

---

> ### Author Rebuttal · Authors · 2025-04-01
>
> We appreciate the detailed feedback and recognition of the clarity and broad applicability of our work. Our response is as follows,
>
>
> **Relationship to other UQ methods.** The reviewer correctly points out that there are other methods for quantifying uncertainty in the context of LLMs besides conformal prediction (CP), including methods for estimating and calibrating confidence. Our goal in this paper was to generate subsets of answer options with guaranteed coverage probability, which makes conformal prediction a natural framework. To the best of our knowledge, no other uncertainty quantification technique provides a similar guarantee. We emphasize that small prediction sets are valuable in and of themselves because they result in lower query costs, so this reduction of the space of answer options is an important feature of our procedure. [Please see the *Small conformal prediction sets reduce costs* section in the response to reviewer `z4Sk`.]
>
> We believe the flexibility in the choice of score function is part of the appeal of CP and CROQ. We agree that it will be important to investigate how CROQ works with other choices of score functions, which we leave for future work.
>
> **Choice of $\alpha$.** Please see the *Choice of $\alpha$* section in our response to reviewer `z4Sk`.
>
> **Theoretical guarantees.** We appreciate the reviewer pointing out that the relationship between Proposition 2.1 and our procedure may not be clear. We have reframed this proposition and its proof slightly to make it clear that our procedure enjoys this coverage guarantee. Regarding the optimization procedure for CP-OPT, we have pointed out (in lines 222-224, second column) that the empirical surrogates converge almost surely to their population counterparts. We have reworded the text to point out that this also holds for the cross-entropy term $\widehat{C}(g)$. We have also added intuition regarding the relationship between problems (P2) and (P1). We defer a more formal convergence analysis for future work.
>
> **Importance and Generality of the MCQ setting.** Please see the same section in our response to the reviewer `p4om`.
>
>
> **Computational cost.** As implemented in our paper, CROQ requires two queries to a given LLM. However, as noted by reviewer `grWo`, it's possible to cache the input such that the difference in cost between one query vs. two will be relatively minimal. Furthermore, as discussed in the *Small conformal prediction sets reduce costs* section in the response to reviewer `z4Sk`, it is possible to use a very cheap method to generate the scores such that the cost (both computational and literal) primarily derives from the query which produces the MCQ answer. For example, the user could use a pre-trained semantic embedding and then use the cosine similarity between the query and each response option as the conformal score. This cost is minimal compared to the cost of an MCQ query, and in general, we expect that it will be more than offset by the reduction in query cost due to the reduction in the number of answer options. Methods that involve self-consistency or self-refinement will, in general, require multiple queries, and therefore, we expect them to be more expensive, but we leave a full investigation for future work. However, we have added a discussion along the lines of the above to the paper appendix.
>
> **Magnitude of CP-OPT gains vs. logits.** While the advantages of CP-OPT over logits in terms of set sizes and accuracy are numerically small, we believe that the real-world difference can be substantial at scale when large numbers of users are querying an LLM repeatedly. The figures and tables in the appendix aim to provide a finer-grained view of when and how CP-OPT improves over logits. For example, we observe that CP-OPT, in general, yields more sets of size 1 than logits and that the accuracies vary as a function of set size (which indicates that set size is a good measure of overall uncertainty). We hypothesize that CP-OPT improves over logits precisely where logits are poorly calibrated, i.e., where an LLM is over- or under-confident. We plan to test this hypothesis in future work.
>
> We hope our response resolves the queries. We are happy to answer any further questions you may have.

---

> > ### Comment · Reviewer_uJFE · 2025-04-03
> >
> > I thank the authors for their responses. However, many of my concerns are still not addressed. For example, the method doesn't compare with other simple prompting methods for MCQ questions and other tool optimization methods for tool selection as baselines given that the improvements with CROQ are not quite significant. Next, the improvements increase as the the number of choices increase. However, adding artificial choices may not reflect natural MCQ distributions and could introduce biases. I will recommend the authors try some datasets with more choices like MMLU Pro. Besides, authors claim that using the cosine similarity between the query and each response option as the conformal score will significantly reduce the cost. But the performance of using this simple score function is doubted.

---

> > > ### Author Response · Authors · 2025-04-09
> > >
> > > Thanks for the response. We provide additional experimental results and clarifications below.
> > >
> > > **On our experimental setup.** We augmented the answer choices by randomly drawing answer choices from other questions, and on MMLU, where questions are labeled with topics, we sampled the additional answer questions from questions on the same topic. Thus, the choices were not arbitrary, and more importantly, the LLMs' baseline accuracy on the datasets with the inflated number of choices decreases substantially. This indicates that the choices introduced are effective distractors.
> > >
> > > **Evaluation on MMLU-pro.** We evaluated CROQ on the MMLU-Pro dataset with questions having 10 options. We observe that the baseline accuracy with the Phi-3 model is 36.4%, and we get a 3% relative improvement in accuracy with CROQ – a significant improvement on a 10-option dataset, particularly given that MMLU-pro contains much harder questions. We will include these results in the paper.
> > >
> > > In addition to this, we have results on a practical application where an agentic system needs to select the right tables for generating SQL for a given natural language query. Please see the response to reviewer `p4om` for details on this.
> > >
> > > **On Prompting Methods.** Our focus and main contributions are on the CROQ procedure that revises the question after pruning options using conformal prediction. For evaluation, we chose a simple and computationally efficient procedure for solving MCQ-type tasks. In this procedure, the forward pass is run on the model to obtain logits (scores) for the answer choices, and the choice with the highest score is selected.
> > >
> > > While CROQ can be used in concert with other prompting strategies, such as CoT, etc., we favored the above procedure for clean and computationally efficient evaluation in contrast to prompting methods, which can involve generating a large number of tokens to get an answer. The generated response also depends heavily on the choice of decoding strategy used. Using CROQ in conjunction with CoT could be an interesting direction for future work.

---

### Official Review · Reviewer_z4Sk · 2025-03-18

**Overall Recommendation:** 3

**Summary:**

This paper is using conformal prediction sets to improve the performance of LLMs on multiple choice question answering tasks. In particular, the propose a framework which they first construct a prediction set and then re-ask the same question with the limited options in the set from the LLM. They then empirically show this method can improve the accuracy of the LLM in a variety of multiple choice tasks. They also, offer some optimization methods to improve the score function used in CP pipeline to promote tighter prediction sets which then improve the effectiveness of the proposed method.


I have read other reviewers comments and i want to keep my score. My major concern, which is remained, is that i still do not understand why CP, its coverage guarantee, and pruning based on CP sets are of practical relevance here. I am very familiar with CP tools, and this does not make sense to me. To push such a narrative, there should be either, a very strong set of experimental setups, where the authors compare with a wide range of pruning techniques and other UQ ideas, or alternatively, a some form of theory, even minimal, that showcases CP sets are the correct tool for this problem.

**Claims And Evidence:**

The claims are clear but the evidence is not entirely convincing. It is not obvious if such a framework can actually improve the accuracy of the LLMs in a meaningful way. In particular, It is not obvious to me how "95 %" sets would be a meaningful notion in this problem, as you will be missing the correct label 5 percents of the time by design, and then forcing the LLM to pick among wrong answers. And then this raises the question of how to pick alpha or whether such a framework makes sense at all or not.

**Essential References Not Discussed:**

Not that I know of.

**Experimental Designs Or Analyses:**

...

**Methods And Evaluation Criteria:**

I do not think the evaluation criteria is sufficient. Due to the concern raised above, I think there should be extra evaluation metrics to have a finer grained understanding of what happens when applying this framework. For instance, it might be the case, even though the overall accuracy is improved, there would a non negligible number of cases that the LLM might have give the correct answer originally, but now due to restricting to a set which does not include the correct label (which happens 5 percents of the time) the LLM is forced to get the wrong answer.

**Other Comments Or Suggestions:**

...

**Other Strengths And Weaknesses:**

My major concern is the lack of any theoretical or even some higher level intuitions/observations/discussions on why such a framework based on CP is the correct way of informing the LLM about its uncertainty. This framework puts a tradeoff between the informativeness of UQ with CP (when choosing alpha too small) vs some inevitable mistakes that we force to LLM (by choosing alpha large), and this does not sound like the correct tradeoff to look at.

For instance, an immediate alternative to the proposed framework could be, we construct the CP sets, but instead of re-asking the question with the limited options, this time we append the set to the context window of the LLM and then explain the correct answer with 95 percents probability is in this set. How does this change the situation? is it better or worse?

**Questions For Authors:**

...

**Relation To Broader Scientific Literature:**

Uncertainty quantification in LLMs is a very important and active field, and the idea of re-prompting the LLM after uncertainty quantification sounds interesting.

**Theoretical Claims:**

There is not much theoretical claims in the paper.

---

> ### Author Rebuttal · Authors · 2025-04-01
>
> We thank the reviewer for careful attention to the paper. Our response to the queries is as follows,
>
> **Choice of $\alpha$.** We set $\alpha$ to a single (fairly arbitrary) value of $0.05$ in the main body of the paper for simplicity of exposition, but $\alpha$ can be treated as a hyperparameter and tuned to maximize accuracy. We discuss this in the Appendix in section B.1, with results in Figures 4 and 5. As discussed in lines 73-83 (first column), we agree that $\alpha$ represents a tradeoff whereby larger values result in smaller sets with some probability of excluding the correct answer. If we set $\alpha = 0$, then we'd include all the answer options each time, which recovers the original MCQ setting, so in some sense, we've simply parameterized a tradeoff that we can then optimize for a downstream criterion of interest like accuracy.
>
> **Evaluation criteria.** We appreciate the reviewer's point. The additional figures and tables in the appendix provide a finer-grained view of (1) the distributions of answer set sizes produced by our procedure, and (2) accuracy conditional on set sizes. In addition, we have added tables that illustrate how often the CROQ procedure causes the LLM to "switch" from incorrect to correct answers and vice versa. We do indeed observe that our CROQ procedure causes the LLM to incorrectly answer some small proportion of questions which it initially answered correctly, but *this is more than offset in general by answers which it initially got incorrect and which it gets correct after CROQ*.
>
> **Small conformal prediction sets reduce costs**. In addition to the goal of improving accuracy, we note that smaller set sizes are generally desirable in and of themselves because fewer tokens in the prompt means lower query costs. This difference can be substantial in settings like text-to-SQL or other tool usage/API selection settings, where the text describing a given answer option can be extremely large. While in our experiments we used the same LLM both to generate the conformal scores for the purposes of constructing the conformal prediction set *and* to generate the final MCQ answers, it's possible to use a model cascade such that a small/cheap model is used to generate the conformal prediction sets and then the MCQ is passed to a larger/more expensive LLM to generate an answer. In ongoing experiments in a text-to-SQL setting, we observe that by using a model cascade like this, we are able to substantially reduce the overall query cost while preserving or improving downstream accuracy. Once again, these tradeoffs can be optimized by tuning $\alpha$. We have added a section to the appendix and a small reference in the main text discussing this. (In addition, please see the existing section B.2 in the appendix for some discussion of model cascades and cost tradeoffs.)
>
> **Why Conformal Prediction (CP) is a useful framework.**  The idea of appending the conformal set to the original query is interesting and we believe would be worth investigating in future work. Our goal with the CROQ procedure however was to *reduce* the amount of uncertainty in MCQ-type queries, rather than to inform the LLM about its own uncertainty per se. We aim to reduce the likelihood that the LLM will be "distracted" by an available incorrect answer by pruning those answers (with high probability). This procedure is motivated by the simple empirical observation in Figure 1 that LLMs are more likely to answer correctly when there are fewer distractor options. Regarding theoretical justification, we emphasize that our procedure satisfies a coverage guarantee (Proposition 2.1), which means that the correct answer will be inadvertently removed at most a proportion $\alpha$ of the time. As discussed above, this is some sense simply a generalization of the vanilla MCQ setting with a parameterized tradeoff that can be optimized.
>
>
> We hope our response resolves the queries. We are happy to answer any further questions.

---

### Official Review · Reviewer_grWo · 2025-03-19

**Overall Recommendation:** 4

**Summary:**

First, the paper observes that removing incorrect answer choices from the answer sets given to an LLM improves performance. This motivates conformal revision of questions (CROQ), a simple method to boost multiple choice QA (MCQA) on any model and any dataset by first asking a question to the model, building a confidence set of MCQA answers that includes the true answer with 1 - $\alpha$ probability, and then re-prompting the model with only the answers in the set as the given answer choices. The confidence sets are built using split conformal prediction.

Next, the paper argues that current prediction logits are not explicitly optimized for producing good confidence sets, and proposes the CP-OPT objective to learn a small auxiliary head off of an LLM that can produce prediction scores that result in more useful (smaller) confidence sets. The equation P1 shows the objective for CP-OPT optimization, which minimizes the expected average confidence set size over the train set, subject to a expected confidence coverage constraint. P2 shows a surrogate objective that relaxes the constraints and relaxes step-wise functions to smooth sigmoids for differentiability. After optimizing with CP-OPT, the hold-out calibration set used for conformal prediction may result in smaller confidence sets at the required level of coverage.

The authors experimentally verify the efficacy of CROQ over standard QA, the generally smaller confidence set size of CP-OPT over logit-based conformal prediction, and the resulting improvements of CROQ when using CP-OPT instead of logits. They also conduct additional ablations and experiments to understand the behavior of the new methods.

**Claims And Evidence:**

* "Our extensive experiments on MMLU, ToolAlpaca, and TruthfulQA datasets with multiple LLMs show that CROQ improves accuracy over the standard inference, with more pronounced gains when paired with CP-OPT." -- well-supported
* The three hypotheses in Section 4 are well-explored, and the evidence given is generally sufficient to support them.
* Other analysis claims are reasonable.

**Essential References Not Discussed:**

Not to my knowledge.

**Experimental Designs Or Analyses:**

I reviewed the main paper experimental designs and the ablations/additional experiments presented in Figures 3-5 in the appendices. I am satisfied with the experimental design, although I do wish a little more attention was given to discussing some of the choices of hyperparameters for the CP-OPT loss objective. See my W3-4 in the Strengths and Weaknesses section.

**Methods And Evaluation Criteria:**

Yes, the datasets are standard QA datasets common for QA tasks. The main metrics (accuracy, coverage, and confidence set size) are suitable for the two objectives of CROQ and CP-OPT, respectively (produce good QA performance and useful confidence sets).

**Other Comments Or Suggestions:**

N/A

**Other Strengths And Weaknesses:**

* S1: Clear presentation and writing
* S2: Motivation is strong
* S3: Results are presented clearly and statistical analysis is presented well
* S4: With proper input caching, it seems the re-prompt step in CROQ could be done rather efficiently, to add relatively little extra inference time.

Weaknesses:
* W1: CP-OPT requires a labeled finetune train step on a particular dataset, while logits do not and can be applied to any QA dataset with only a calibration step to select the conformal prediction threshold.
* W2: There is little analysis on how a tuned CP-OPT head for a particular dataset/model might be used/generalized to other settings (with merely a new calibration step) to avoid retraining. I find it unlikely that it would work well between models, as the activations are different, but I wonder if it would work well between datasets.
* W3: In P2, there are two different trade-off weight hyperparameters, lambda and lambda_1. I only see discussion of lambda in Appendix E. It's not clear how lambda_1 is set.
* W4: The lambda parameter describes how strongly to consider the soft coverage constraints while training the CP-OPT model. In Appendix E, we see that the lambda choices for different model/dataset settings are widely different. For example, in some cases, the chosen lambda is 0.1, and in other cases it is 10. This means a few things: first, there is some additional cost and effort required to tune this hyperparameter compared to the logits procedure, which has no such hyperparameter. Second, it is unclear how this hyperparameter was tuned--was it trained with signal on the train set only? the train and calibration sets?

**Questions For Authors:**

* Q1: In equation 2, is there possibly a slight definitional issue? It seems to me that the verbal description of the threshold as "the smallest empirical quantile of the scores for the correct answers on the calibration dataset that is sufficient to satisfy (an empirical version of) the coverage property" does not quite match with the definition of the conformal sets as being everything with score greater than or equal to the threshold -- do we not then want the "largest empirical quantile that is sufficient to satisfy the coverage property?" And would the proper equation then be something more like a max {q} such that some empirical fraction of examples at or above the threshold is greater than or equal to 1 - alpha?
* Q2: The tradeoff between coverage level of the pruned set and boosted performance in the revised task seems interesting to draw some conclusion about in a very simplified toy setting. With some strong toy assumptions about the behavior of the relationship depicted in Figure 1 (i.e. choosing a closed-form function that acts something like the monotonic curves shown in Figure 1), could we make any argument about what the ideal alpha would be to minimize the tradeoff between missed answers in the confidence set and improved accuracy in the second QA step? I do not think this analysis is critical for inclusion in this paper, but I am interested in it!

**Relation To Broader Scientific Literature:**

While prior works (Vovk et al., 2005; Angelopoulos et al., 2022, ) use conformal prediction as a way of expressing the confidence of machine learning systems in predictions in the form of calibrated-size prediction confidence sets, this seems to my knowledge the first work to use these confidence sets to boost MCQA by iteratively narrowing down the answer choices used to prompt a model.

There is a large literature of producing calibrated and accurate confidence scores along with LLM QA predictions (Tian et al. 2022, Kadavath et al. 2022, Sebastian et al. 2024). These methods rely on consistency of sampling, logit values, calibrated confidence prediction heads, and textual elicitation to compute confidence scores associated with a final answer. This is a bit of a different motivation from the conformal prediction, but employing a similar scope of techniques.

Farquhar, Sebastian, et al. "Detecting hallucinations in large language models using semantic entropy." Nature 630.8017 (2024): 625-630.

Kadavath, Saurav, et al. "Language models (mostly) know what they know." arXiv preprint arXiv:2207.05221 (2022).

Tian, Katherine, et al. "Just ask for calibration: Strategies for eliciting calibrated confidence scores from language models fine-tuned with human feedback." arXiv preprint arXiv:2305.14975 (2023).

Other citations are present in the paper's bibliography.

**Theoretical Claims:**

There are not any proofs (mostly, the paper relies on theoretical arguments from other works).

---

> ### Author Rebuttal · Authors · 2025-04-01
>
> We appreciate the thoughtful review and the noted strengths on presentation, problem motivation, empirical evaluation, and results of our paper. Our response to the queries and comments is as follows.
>
> **Input caching to make re-prompting efficient.** We thank the reviewer for the suggestion to use input caching to reduce the inference time in the second round of CROQ. Since the part of the question without the options remains the same, we can definitely cache the inference output on tokens in the question and re-use this in the next round's inference with a reduced set of options. We plan to implement and release this soon.
>
>
> **CP-OPT requires training a small neural network on a dataset.**  As noted, logit scores can be obtained off-the-shelf from the language model. However, they can be unreliable and could be less effective for downstream use cases such as ours. To improve the quality of the scores for the application at hand, it might be necessary to tune the scores accordingly. Note that our procedure CP-OPT to tune scores is light-weight — it only requires training a small 2-layer neural network $g$ on features extracted from the LLM, and the inference overhead of $g$ is negligible compared to the LLM inference which is also needed for logit scores. Thus, while there is an extra compute cost with CP-OPT, it is insignificant in comparison to the LLM inference cost, and this cost pays off with improved performance. It also pays off with reduced set sizes, which means lower query costs. Please see the paragraph *Small conformal prediction sets reduce costs* in our response to reviewer `z4Sk`.
>
>
> **Re-use $g$ across models and datasets.** We appreciate the thoughts to re-use the $g$ to mitigate this minor cost of re-training it for different datasets and models. On sharing $g$ across LLMs, we agree with your view on this. Since $g$ is trained on features on features from an LLM, using it on features from another LLM will likely not work. Moreover, different datasets may differ in features and thus a $g$ trained on one dataset may not work well on another dataset. We make two final remarks on this,
>
> i) The cost of training $g$ is insignificant compared to the LLM compute costs.
>
> ii) Reuse of $g$ could be achieved for instance by incorporating multiple models and datasets while training $g$ or first learning some model invariant representations that are fed to $g$. Exploring this could be an interesting future work.
>
> **W3.** $\lambda_1$ corresponds to the weight decay hyperparameter.
>
> **Discussion on choices of hyperparameters for CP-OPT.** Yes, there is a small cost of tuning the hyperparameters. We select the hyperparameters by observing the performance on the validation data. We have included a more detailed discussion on this in the paper.
>
> **Equation (2) clarification.** Thanks for your careful attention. We believe there is some confusion due to the interpretation of scores. Some of the canonical works (Angelopoulos & Bates, 2022) in conformal prediction have used *non-conformity scores* for $g$, i.e., lower is better. In our work, we interpret $g$ as measuring *conformity*, i.e., higher is better. Thus, in our setting, the threshold will be the lowest $\alpha$ quantile of the scores, hence equation (2).
>
> **Analysis of CROQ in toy setting.** This is indeed very intriguing. We appreciate the reviewer's enthusiasm. We can characterize the accuracy gain and $\alpha$ if we assume the LLM satisfies *monotone accuracy property* as in Figure 1.
>
> Consider a predictor (LLM) that has accuracy $f(k)$ on questions with $k$ choices. It is fair to assume that as the number of choices $k$ decreases, the accuracy $f(k)$ increases, i.e. $f$ is a monotonically decreasing function of $k$. This is also confirmed in our experiments (Figure 1). We refer to this as the monotone accuracy property of the predictor.
>
> Now, let the initial number of options in the questions be $M$ and after revising them with conformal prediction (CP) the questions have $m<M$ choices and it is guaranteed by CP that the true answer is still in the $m$ choices for $1-\alpha$ fraction of the questions. Then, the gain in accuracy after CROQ is as follows,
>
> $$\text{Gain} = \text{Accuracy After} - \text{Accuracy Before}$$ The $\text{Accuracy After} = f(m)$ times the fraction of questions for which true choice is in the revised question = $f(m)(1-\alpha)$
>
> $$\Delta(M,m,\alpha) = f(m)(1-\alpha) - f(M) = f(m) - f(M) - \alpha f(m)$$. Now we can make two claims,
>
> * If $\alpha$ is fixed, then we should see improvements whenever $f(m) > \frac{f(M)}{1-\alpha}$.
>
> * If $\alpha$ is not fixed, then the gain $\Delta(M,m,\alpha) > 0$, for any $\alpha < \frac{f(m) - f(M)}{f(m)}$. By the monotone accuracy property of the predictor $f(m) - f(M) >0$, that means any $\alpha \in (0, \frac{f(m) - f(M))}{f(m)})$ will yield a gain in accuracy.
>
> -----
>
> We hope our response resolves the queries. We are happy to answer any further questions.

---

> > ### Comment · Reviewer_grWo · 2025-04-08
> >
> > Thank you, I appreciate these responses and the additional theoretical treatment. I maintain my score and recommend acceptance. I somewhat disagree with Reviewer p4om who mentions limited scope of experiments -- in fact I find the QA experiments very reasonable, and believe agentic experiments might muddy the clean evaluation of this method against other QA methods.
> >
> > Re: Equation 2, I guess what I should say is that even if one does consider a conformity score, I think the minimum operator as written results in a set which _does not_ satisfy the empirical coverage property, as the empirical miscoverage rate, I imagine, should really be smaller than alpha. Do you disagree?

---

> > > ### Author Response · Authors · 2025-04-09
> > >
> > > Thanks for the reply and endorsing our empirical evaluation.
> > >
> > > **Equation 2 correction.** Thank you for raising the query on equation 2. The min and direction of the inequalities are still correct. However, we realized upon review that we were missing a correction factor, which, when included, could indeed result in the empirical miscoverage rate being smaller than $\alpha$. While the correction factor can be inserted into the current equation (2) expression, for increased clarity, we have rewritten the definition of the threshold ```{\hat{\tau}_\alpha}``` as the ```\lfloor (n+1)*\alpha \rfloor / n}```  empirical quantile of the scores from the calibration dataset. This gives the desired coverage guarantee, following the same proof technique that is in Appendix D of Angelopoulos & Bates (2022), "A Gentle Introduction to Conformal Prediction". Using their notation, we have ```\hat{\tau}_{\alpha} = s_{\lfloor (n+1)*\alpha \rfloor}```, and for any test point $(X_\text{test}, Y_\text{test})$ with corresponding conformal score $s_\text{test}$ and prediction set $\mathcal{C}(X_\text{test})$:
> > >
> > > ``` \mathbb{P}(Y_\text{test} \in \mathcal{C}(X_\text{test})) = \mathbb{P}(s_\text{test} \geq s_{\lfloor (n+1)*\alpha \rfloor}) = (n - \lfloor (n+1)*\alpha \rfloor + 1)/(n+1) \geq 1 - \alpha ```.
> > >
> > > We have included this proof in the appendix of the paper and slightly modified the text accordingly.

---

### Official Review · Reviewer_p4om · 2025-03-27

**Overall Recommendation:** 2

**Summary:**

This paper proposes a method for improving the performance of LLMs on MCQ benchmarks using conformal prediction. The key idea is to construct an uncertainty set that contains the correct answer with high probability, prune all answer choices that are outside the set, and then present the LLM with a reduced set of choices that only lie in the set. The motivation for the paper seems to be that zero-shot performance of LLMs degrade on MCQ benchmarks as the answer choices increase. The authors show that this conformal-guided approach marginally improves accuracy of LLMs on some commonly used MCQ benchmarks.

**Claims And Evidence:**

**Main claim:** Statistical uncertainty measures can be used to reduce the set of options in MCQ benchmark items, which in turn can improve the accuracy of the LLM when operating on a pruned set of choices instead of the full set.

**Evidence:** Improvements in accuracy of LLMs on the MMLU, ToolAlpaca, and TruthfulQA benchmarks.

**Essential References Not Discussed:**

The paper covered two strands of literature: applications of conformal prediction to LLMs, and "optimizing" conformal prediction procedure. I think the paper is missing a discussion of other approaches for improving the zero-shot performance of LLMs in MCQ benchmarks, e.g. chain of thought prompting, etc. I think this is important since the goal of the paper is to improve accuracy and not quantify predictive uncertainty.

**Experimental Designs Or Analyses:**

I checked the soundness of the experiments and I think the evaluation procedure is sensible though there is a lack of baselines for improving zero-shot MCQ performance of LLMs using methods other than uncertainty-based pruning.

**Methods And Evaluation Criteria:**

The paper uses standard datasets and evaluation metrics. However, the paper lacks any baselines apart from ablations of the proposed method.

**Other Comments Or Suggestions:**

Given the marginal gains in accuracy, I think it is important to consider other prompting-based baselines for improving the accuracy of LLMs to evaluate of the gains from pruning-based method vs. prompting-based methods.

**Other Strengths And Weaknesses:**

My primary concern with this submission is that it does not address a meaningful problem. MCQ benchmarks are designed to evaluate the capabilities of LLMs in various areas, such as reasoning abilities and knowledge comprehension. However, the MCQ format itself does not represent a meaningful real-world task or practical goal. Consequently, the proposed method appears to exploit the artificial structure of MCQ benchmarks rather than genuinely enhancing the underlying capabilities of the LLM.

Although the paper motivates its proposed method by emphasizing the need for accurate decision-making, this method is specifically tailored to MCQ-formatted tasks, which are not representative of real-world decision-making scenarios. Furthermore, the improvements in accuracy reported in Tables 2 and 3 appear marginal and only become noticeable when the number of choices approaches 15. Given that the methods used to construct the sets are themselves not novel, these factors collectively make the potential impact and contribution of the paper unclear.

**Questions For Authors:**

- If you are using a standard split conformal procedure, does this mean your sets are not adaptive to the conditional uncertainty given the prompt?

- What realistic decision-making scenarios this method can be applies for?

**Relation To Broader Scientific Literature:**

The paper does not include new contributions in conformal prediction methodology. The backbone of the proposed method is a simple split conformal procedure, and the CP-OPT method is largely based on (Stutz et al., 2022), but with application to post-hoc features instead of end-to-end training of the LLM. The key contribution of the paper is an ad-hoc way to revise the LLM answers to MCQ questions by applying existing conformal prediction methods to construct prediction sets that filter out unlikely answers.

**Theoretical Claims:**

The theoretical validity claim in Proposition 2.1 is a known result in the conformal prediction literature. No other theoretical claims are made by the authors.

---

> ### Author Rebuttal · Authors · 2025-04-01
>
> Thanks for the feedback. Our response to the queries is as follows,
>
> **Importance and generality of the MCQ setting.** The multi-choice question-answering framework encompasses any setting in which an LLM must select from among a finite number of options. We believe that this describes many if not most steps in `agentic workflows`. This includes selecting function calls or APIs, selecting among UI elements, selecting databases or tables in text-to-SQL settings, selecting which agent to pass an output to next, selecting the next step in a plan, selecting a document from a database for RAG, selecting a set of in-context examples from a database of such examples, selecting diagnosis codes or labels, etc. The answer options do not necessarily need to be defined in advance: in open-ended response settings, for example, it's possible for example to have an LLM generate a shortlist of initial candidates which can then be pruned before being passed to other agents. We agree that it is an interesting open question how to extend our framework to the open-ended response setting, but we believe that the MCQ abstraction is widely applicable on its own.
>
> **Comparison with prompting-based methods.** Our work's focus is to evaluate the hypothesis that conformal prediction based pruning can be effective in improving accuracy. Comparison with prompting-based methods would be interesting future work.
>
> **Other advantages of pruning answer choices.** In addition to improving accuracy, pruning answer choices can lead to computational and dollar savings. This can occur when a score function is used in the conformal procedure that is cheap to compute relative to the cost of the MCQ query. For more details, please see the section *Small conformal prediction sets reduce costs* in our response to reviewer `z4Sk`, and the section *Computational cost* in our response to reviewer `uJFE`.
>
> **On conditional uncertainty given the prompt:** In general conditional calibration is not possible in conformal prediction (Barber et al. 2020). Due to this limitation, conformal prediction with marginal coverage guarantee is widely used and it is suitable for our work where we aim to evaluate the hypothesis that conformal prediction based pruning can be effective in improving accuracy.
>
> Further, the logit scores are generated by running the forward pass of the LLM on the entire prompt including all the answer options; these scores then serve as features, along with the other features described, for the CP-OPT procedure. In that sense, both the logit scores and our CP-OPT scores represent the uncertainty conditional on the prompt, and the resulting sets reflect this uncertainty.
>
>
> **Regarding realistic decision-making scenarios**, our method can be applied to any setting in which an LLM must choose from among a finite set of response options, whether those options represent question answers, APIs, function calls, downstream agents which can receive input, etc. We refer the reviewer to the section *Importance and generality of the MCQ setting* above.
>
> We hope our response resolves the queries. We are happy to answer any further questions you may have.

---

> > ### Comment · Reviewer_p4om · 2025-04-07
> >
> > Thanks for the responses. It could be interesting to apply this method in agentic workflows to see if it improves the availability of LLMs to pick the right actions and selection in context. I think that this work would benefit from fleshing out the real-world applications a little by including such agentic benchmarks. Currently, the experiments only show very marginal gains with no strong baselines and only in contrived MCQ settings, which makes it hard to judge the significance of the contribution.

---

> > > ### Author Response · Authors · 2025-04-09
> > >
> > > For an application in an agentic workflow, we consider the Natural Language Question to SQL (NL2SQL) task, where an LLM-based agent generates a SQL query for a user's natural language question. A component of the standard agentic workflow in this task is to first predict the relevant tables whose schema should be included in the context of the LLM, which generates the SQL query. This step is critical to decrease cost and, in some cases, is necessary when the full database schema would exceed the LLM's context limit.
> > >
> > > We consider the BIRD dataset (https://arxiv.org/pdf/2305.03111) - a large benchmark that contains 12,751 NLQ-SQL pairs across 95 databases. We filter out databases with 20 tables or more  (to avoid context limit errors) and remove the retail_world databases due to inconsistent table naming. We considered the following settings:
> > >
> > > **Approach 1** - Include all table schemas in the LLM prompt.
> > >
> > > **Approach 2** - Include all table schemas for tables whose cosine similarity score is greater than a particular threshold, up to a maximum of 10 tables. The cosine similarity is taken between the embeddings of the natural language question and the table name using the OpenAI text-embedding-ada-002 model. Coverage is defined to include all tables used in the annotated ground-truth SQL query. Coverage was approximately 90%, although this was not explicitly controlled.
> > >
> > > **Approach 3** - Include tables selected using conformal prediction (CP) on CP-OPT scores. This is equivalent to the CROQ procedure, where the scores for CP are obtained from a source other than LLM. More specifically, we learn CP-OPT scores using embeddings of natural language questions and table names.
> > >
> > >
> > >
> > > We used 3412 NLQ-SQL pairs for training in approach 3, and validated on 3411 examples in approach 2 and 3. We then tested the 3 approaches on 200 NLQ-SQL pairs. We use GPT4-0613 as the LLM for SQL query generation, and report the execution accuracy, average set size, and total token cost. The results in all three settings are summarized in the table below,
> > >
> > > |           | Accuracy | Avg. Set Size | Coverage | LLM Cost |
> > > | --------  | -------- | --------      | -------- | -------- |
> > > | Approach 1 | 32.0%    | 7.270         | 100%     | $7.10    |
> > > | Approach 2 | 29.5%    | 6.405         | 88%      | $6.63    |
> > > | Approach 3 (Ours) | **32.5%**    | **2.685**         | 92%      | **$3.89**   |
> > >
> > > Here, the set size means the number of tables whose schema will be included in the LLM context. Thus, lower avg. set size means fewer tables (and hence fewer tokens) in the LLM context. In the results, we see a significant reduction in the avg. set size in approach 3 while maintaining high coverage (92%). This results in a substantial reduction in the number of tokens in the LLM context, **leading to a 45% decrease in LLM cost** all while achieving slightly higher accuracy in comparison to approach 1.

---

### Decision · Program_Chairs · 2025-05-01

**Decision:**

Accept (poster)

**Comment:**

This paper considers the problem of improving the accuracy of LLMs on multiple choice questions by reducing the options using conformal prediction based uncertainty sets. Conformal training based approach is used to optimize the set size to achieve high coverage (e.g., 95%). The paper shows that this conformal predication based pruning approach results in accuracy improvements of varying degrees.

The reviewers' for this paper were split (three negative and two positive) before the discussion. The reviewers' raised a number of questions and concerns. Authors' rebuttal addressed some of them satisfactorily. During the AC-reviewer discussion, we discussed a number of points.
1. All of them acknoqledged that technical novelty is low. The positive reviewers' argued that the simplicity is a strength and this idea hasn't been explored before.
2. Marginal improvement in performance. The positive reviewers' pointed out the statistical significance of the results.
3. Lack of strong baselines for MCQ approaches. Authors' provided some additional experiments in their rebuttal.
4. The proposed approach lacks principled justification. Authors' provided "Analysis of CROQ in toy setting" in the rebuttal to address this concern.

One of the reviewer who changed their score from 2 to 3 mentioned the following for point #4: "Under the condition of including a more formal version of this toy argument in the revised version, and communicating clearly that there is a need for more theoretical efforts to justify the CP pruning, I am increasing my score to 3."

Based on the overall discussion, reviewers' are leaning to accept the paper if point #4 is addressed in the final paper. Therefore, I recommend accepting this paper and strongly encourage the authors' to incorporate all the discussion in the camera copy to further improve the paper.